# MedVAE: Efficient Automated Interpretation of Medical Images with Large-Scale Generalizable Autoencoders

| | |
|---|---|
| **Maya Varma**[*][1] | MAYAVARMA@CS.STANFORD.EDU |
| **Ashwin Kumar**[*][1] | AKKUMAR@STANFORD.EDU |
| **Rogier van der Sluijs**[*][1] | SLUIJS@STANFORD.EDU |
| **Sophie Ostmeier**[1] | SOSTM@STANFORD.EDU |
| **Louis Blankemeier**[1] | LBLANKEM@STANFORD.EDU |
| **Pierre Chambon**[1] | PCHAMBON@STANFORD.EDU |
| **Christian Bluethgen**[1] | BLUETHGEN@STANFORD.EDU |
| **Jip Prince**[2] | JIPFPRINCE@GMAIL.COM |
| **Curtis Langlotz**[1] | LANGLOTZ@STANFORD.EDU |
| **Akshay Chaudhari**[1] | AKSHAYSC@STANFORD.EDU |

[1] *Stanford Center for Artificial Intelligence in Medicine and Imaging, Stanford University, USA*
[2] *UMC Utrecht, Netherlands*

**Editors:** Accepted for publication at MIDL 2025

## Abstract

Medical images are acquired at high resolutions with large fields of view in order to capture fine-grained features necessary for clinical decision-making. Consequently, training deep learning models on medical images can incur large computational costs. In this work, we address the challenge of downsizing medical images in order to improve downstream computational efficiency while preserving clinically-relevant features. We introduce *MedVAE*, a family of six large-scale 2D and 3D autoencoders capable of encoding medical images as downsized latent representations and decoding latent representations back to high-resolution images. We train MedVAE autoencoders using a novel two-stage training approach with 1,052,730 medical images. Across diverse tasks obtained from 20 medical image datasets, we demonstrate that (1) utilizing MedVAE latent representations in place of high-resolution images when training downstream models can lead to efficiency benefits (up to 70x improvement in throughput) while simultaneously preserving clinically-relevant features and (2) MedVAE can decode latent representations back to high-resolution images with high fidelity. Our work demonstrates that large-scale, generalizable autoencoders can help address critical efficiency challenges in the medical domain.[1]

**Keywords:** computer-aided detection and diagnosis, variational autoencoders, efficiency

## 1. Introduction

Medical images (e.g. X-rays, computed tomography (CT) scans) are essential diagnostic tools in clinical practice. Since medical conditions are often characterized by the presence of subtle features, images are generally acquired with high spatial resolution and large fields of view in order to capture the required level of diagnostic detail for interpretation by radiologists (Huda and Abrahams, 2015). However, high-resolution medical images,

---

[*] Equal Contribution

1. Code: https://github.com/StanfordMIMI/MedVAE

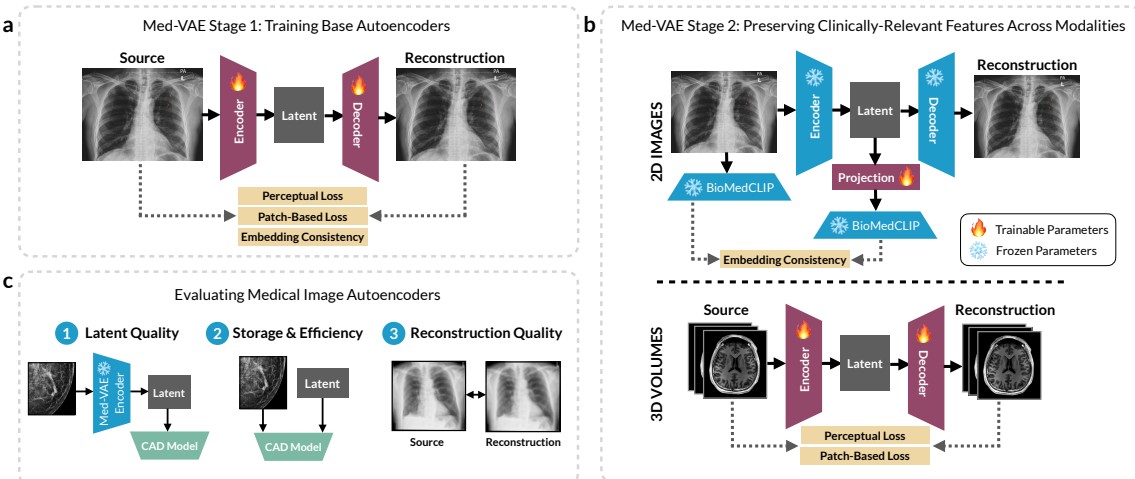

Figure 1: We introduce MedVAE, a suite of large-scale autoencoders capable of downsizing medical images to latent representations and decoding latent representations back to images.

especially volumetric (3D) images, can result in large data storage costs and increased or even intractable computational complexity for downstream computer-aided diagnosis (CAD) models (Freire et al., 2022; Tan and Le, 2019). This is likely to become a significant concern in the near future due to the rapid growth of medical imaging volumes stored by hospitals (Mesterhazy et al., 2020), the expanding use of CAD tools in clinics (Dikici et al., 2020; Najjar, 2023), and paradigm shifts towards large-scale foundation models (Bommasani et al., 2022; Chen et al., 2024; Blankemeier et al., 2024). Many existing CAD models address this challenge by interpolating images to lower resolutions, despite the lower performance of models trained on interpolated data (Sabottke and Spieler, 2020a; Huang et al., 2023).

A promising solution lies in powerful autoencoder methods, which are capable of encoding images as downsized latent representations and decoding latent representations back to images. Recent works, particularly in the context of latent diffusion models, have demonstrated that downsized latent representations can capture relevant spatial structure from high-resolution input images while simultaneously improving efficiency on tasks such as image generation (Rombach et al., 2022). These findings suggest that autoencoders may hold potential for addressing the aforementioned storage and efficiency challenges in the medical domain by encoding high-resolution images as downsized latent representations, which can be used to develop downstream CAD models at a fraction of the computational cost.

Several large-scale autoencoders have been introduced in recent years (Rombach et al., 2022; Lee et al., 2023); however, directly applying these models to the medical domain is challenging since medical images include a diverse range of clinically-relevant features (e.g. tumors, lesions, fractures), anatomical regions of focus (e.g. head, chest, knee), and modalities (e.g. 2D and 3D images). An effective generalizable autoencoding approach in the medical image domain must operate across a wide range of medical images and preserve clinically relevant features in both downsized latents as well as decoded reconstructions. However, existing autoencoder models are either (a) developed for natural images (Rombach et al., 2022), which represent a significant domain shift from medical images, or (b)

developed for a focused set of medical images (e.g. chest X-rays) (Lee et al., 2023) and are not explicitly trained to preserve clinically-relevant features across diverse medical images.

In this work, we address these limitations by introducing MedVAE, a family of 6 large-scale, generalizable 2D and 3D autoencoder models developed for the medical image domain. We first curate a large-scale training dataset with over one million 2D and 3D images, and we perform model training using a novel two-stage training scheme designed to optimize quality of latent representations and decoded reconstructions.

We evaluate the quality of latent representations (using 8 CAD tasks) and reconstructed images (using both automated and manual perceptual quality evaluations) with respect to the preservation of clinically-relevant features. Evaluations are derived from 20 multi-institutional, open-source medical datasets with 4 imaging modalities (X-ray, full-field digital mammograms, CT, and MRI) and 8 anatomical regions. We measure the extent to which MedVAE latent representations and reconstructed images can contribute to downstream storage and efficiency benefits while simultaneously preserving clinically-relevant features. Ultimately, our results demonstrate that (1) downsized MedVAE latent representations can be used as drop-in replacements for high-resolution images in CAD pipelines while maintaining or exceeding performance; (2) downsized latent representations reduce storage requirements (up to 512x) and improve downstream efficiency of CAD model training (up to 70x in model throughput) when compared to high-resolution input images; and (3) decoded reconstructions effectively preserve clinically-relevant features as verified by an expert reader study. Our results also demonstrate that MedVAE models outperform existing natural image autoencoders.

Ultimately, we demonstrate the potential that large-scale, generalizable autoencoders hold in addressing the critical storage and efficiency challenges currently faced by the medical domain. Utilizing MedVAE latent representations instead of high-resolution images in training pipelines can improve model efficiency while preserving clinically-relevant features.

## 2. Related Work

Prior computational models trained on high-resolution medical images generally downsize images using interpolation methods, which are provided as a part of standard machine learning packages including PyTorch and OpenCV. However, such approaches have been shown to result in degraded performance, raising the possibility of using learned approaches, such as autoencoders (Sabottke and Spieler, 2020b). Autoencoders are classic machine learning models consisting of encoders, which encode input images as downsized latent representations, and decoders, which decode latent representations back to the pixel space (Kingma and Welling, 2013; He et al., 2021; Vincent et al., 2008). In this work, we specifically focus on autoencoders that generate structured latent representations interpretable as downsized images. We note here that many standard autoencoders instead yield latent representations that take the form of vectors (He et al., 2021; Zhou et al., 2023); we consider these models out of scope for this work, since resulting latents cannot be used as drop-in replacements for images in CAD pipelines and will require downstream architecture modifications (particularly for fully-convolutional CAD models). Additionally as mentioned in the introduction (Section 1), it is unclear whether existing natural image autoencoders (Rombach et al., 2022; Kingma and Welling, 2013; Esser et al., 2021; Krasin et al., 2017) can effectively capture

fine-grained clinically-relevant features. Therefore, in our work, we focus on developing and evaluating large-scale, generalizable autoencoders capable of operating on diverse features, anatomical regions, and modalities.

## 3. Methods

We now present our approach for training generalizable autoencoders for the medical image domain. Autoencoding methods are capable of encoding high-resolution images as downsized latent representations. For a given 2D input image with dimensions $H \times W$ with $B$ channels, an autoencoding method will output a downsized latent representation of size $H/(\sqrt{f}) \times (W/\sqrt{f}) \times C$. Here, $f$ represents the downsizing factor applied to the 2D area of the image and $C$ represents a pre-specified number of latent channels. 3D autoencoding methods follow a similar formulation, where input images are 3D in nature with dimensions $H \times W \times S$ with $B$ channels. Here, the downsizing factor $f$ is applied to the 3D volume of the image; as a result, the latent representation will have dimensions $(H/(\sqrt[3]{f}) \times (W/\sqrt[3]{f}) \times (S/(\sqrt[3]{f}) \times C$. Autoencoding methods are also capable of decoding latent representations back to reconstructed high-resolution images.

We aim to develop large-scale, generalizable medical image autoencoders capable of preserving diverse clinically-relevant features in both latent representations and reconstructions. To this end, we first collect a large-scale training dataset with 1,021,356 2D images and 31,374 3D images curated from 19 multi-institutional, open-source datasets (Johnson et al., 2019; Feng et al., 2021; Jeong et al., 2022; Sorkhei et al., 2021; RSNA, 2023; Nguyen et al., 2022; Moreira et al., 2012; Cai et al., 2023; Jack Jr et al., 2008; Dagley et al., 2017; Insel et al., 2020; LaMontagne et al., 2019; Bien et al., 2018; Hooper et al., 2021; Chilamkurthy et al., 2018; Wasserthal et al., 2023; Ji et al., 2022; Armato III et al., 2011; Stanford Center for Artificial Intelligence in Medicine & Imaging (AIMI), 2024). Images are obtained from two chest X-ray datasets, six full-field digital mammogram (FFDM) datasets, four T1- and T2-weighted head magnetic resonance imaging (MRI) datasets, one knee MRI dataset, two head/neck CT datasts, two whole-body CT datasets, and two chest CT datasets.

We utilize this dataset to train a family of generalizable autoencoders for medical images. Motivated by prior work on natural images (Rombach et al., 2022), we utilize variational autoencoders (VAEs) as the model backbone. We perform model training using a novel two-stage training scheme designed to optimize quality of latent representations and decoded reconstructions. Specifically, the first stage involves training base autoencoders using 2D images (Fig. 1a); we maximize the perceptual similarity between input images and reconstructed images using a perceptual loss (Zhang et al., 2018), a patch-based adversarial objective (Isola et al., 2018), and a domain-specific embedding consistency loss. Whereas existing works on autoencoders train using only this stage, the medical image domain introduces the added complexity of subtle, fine-grained features required for clinical interpretation; thus, we introduce a second stage of training, which aims to further refine the latent space such that clinically-relevant features are preserved across various modalities (Fig. 1b). Specifically, in the context of 2D modalities (e.g. X-ray, FFDM), the second training stage leverages the embedding space of BiomedCLIP, a recently-developed medical foundation model (Zhang et al., 2023), to enforce feature consistency between input images and latent representations. In the context of 3D modalities (e.g. CT, MRI), the second

| Method | $f$ | $C$ | AUROC ↑ | | | | | |
|---|---|---|---|---|---|---|---|---|
| | | | Malignancy (FFDM) | Calcification (FFDM) | BI-RADS (FFDM) | Bone Age (X-ray) | Wrist Fracture (X-ray) | Average |
| High-Resolution | 1 | 1 | $\mathbf{66.1}_{\pm 0.5}$ | $\mathbf{62.4}_{\pm 0.6}$ | $\mathbf{63.4}_{\pm 0.1}$ | $\mathbf{80.2}_{\pm 0.1}$ | $\mathbf{73.7}_{\pm 0.0}$ | $\mathbf{69.2}$ |
| Nearest | 16 | 1 | $65.5_{\pm 0.1}$ | $59.7_{\pm 0.3}$ | $62.4_{\pm 0.1}$ | $81.6_{\pm 0.1}$ | $70.5_{\pm 0.0}$ | $67.9$ |
| Bilinear | 16 | 1 | $65.5_{\pm 0.1}$ | $58.1_{\pm 0.3}$ | $61.1_{\pm 0.2}$ | $81.6_{\pm 0.0}$ | $\mathbf{71.2}_{\pm 0.1}$ | $67.5$ |
| Bicubic | 16 | 1 | $65.5_{\pm 0.4}$ | $58.5_{\pm 0.5}$ | $61.1_{\pm 0.0}$ | $81.8_{\pm 0.2}$ | $71.1_{\pm 0.1}$ | $67.6$ |
| KL-VAE | 16 | 3 | $59.7_{\pm 0.2}$ | $59.1_{\pm 0.3}$ | $58.5_{\pm 0.1}$ | $74.3_{\pm 0.1}$ | $64.5_{\pm 0.1}$ | $63.2$ |
| VQ-GAN | 16 | 3 | $57.4_{\pm 0.3}$ | $58.2_{\pm 0.4}$ | $62.3_{\pm 0.1}$ | $79.1_{\pm 0.2}$ | $65.8_{\pm 0.1}$ | $64.6$ |
| 2D MedVAE | 16 | 1 | $63.6_{\pm 0.6}$ | $\mathbf{63.9}_{\pm 0.4}$ | $\mathbf{65.3}_{\pm 0.2}$ | $\mathbf{84.6}_{\pm 0.1}$ | $70.3_{\pm 0.1}$ | $\mathbf{69.5}$ |
| 2D MedVAE | 16 | 3 | $\mathbf{66.1}_{\pm 0.2}$ | $61.7_{\pm 0.2}$ | $62.3_{\pm 0.1}$ | $82.1_{\pm 0.1}$ | $70.6_{\pm 0.1}$ | $68.6$ |
| Nearest | 64 | 1 | $63.0_{\pm 0.1}$ | $58.8_{\pm 0.2}$ | $60.0_{\pm 0.2}$ | $72.1_{\pm 0.0}$ | $65.1_{\pm 0.1}$ | $63.8$ |
| Bilinear | 64 | 1 | $61.5_{\pm 0.3}$ | $56.9_{\pm 0.4}$ | $\mathbf{61.3}_{\pm 0.1}$ | $72.8_{\pm 0.5}$ | $\mathbf{67.9}_{\pm 0.1}$ | $64.1$ |
| Bicubic | 64 | 1 | $61.2_{\pm 0.5}$ | $57.6_{\pm 0.4}$ | $61.1_{\pm 0.1}$ | $72.8_{\pm 0.2}$ | $67.9_{\pm 0.2}$ | $64.1$ |
| KL-VAE | 64 | 4 | $62.2_{\pm 0.7}$ | $55.8_{\pm 0.4}$ | $56.8_{\pm 0.1}$ | $65.7_{\pm 0.0}$ | $58.8_{\pm 0.0}$ | $59.9$ |
| VQ-GAN | 64 | 4 | $64.5_{\pm 0.5}$ | $57.3_{\pm 0.3}$ | $56.6_{\pm 0.1}$ | $67.6_{\pm 0.1}$ | $61.6_{\pm 0.2}$ | $61.5$ |
| 2D MedVAE | 64 | 1 | $59.0_{\pm 0.3}$ | $\mathbf{59.4}_{\pm 0.7}$ | $60.7_{\pm 0.1}$ | $\mathbf{73.5}_{\pm 0.2}$ | $64.3_{\pm 0.1}$ | $63.4$ |
| 2D MedVAE | 64 | 4 | $\mathbf{64.9}_{\pm 0.2}$ | $58.5_{\pm 0.3}$ | $60.6_{\pm 0.0}$ | $73.0_{\pm 0.2}$ | $66.7_{\pm 0.1}$ | $\mathbf{64.7}$ |

| Method | $f$ | $C$ | AUROC ↑ | | | |
|---|---|---|---|---|---|---|
| | | | Spine Fractures (CT) | Skull Fractures (CT) | Knee Injury (MRI) | Average |
| High-Resolution | 1 | 1 | $\mathbf{82.9}_{\pm 2.2}$ | $\mathbf{63.9}_{\pm 6.3}$ | $\mathbf{69.9}_{\pm 0.6}$ | $\mathbf{72.2}$ |
| Bicubic | 64 | 1 | $77.3_{\pm 4.1}$ | $64.8_{\pm 4.0}$ | $66.4_{\pm 2.3}$ | $69.5$ |
| KL-VAE | 64 | 3 | $68.8_{\pm 2.1}$ | $40.7_{\pm 9.1}$ | $63.9_{\pm 8.2}$ | $57.8$ |
| VQ-GAN | 64 | 3 | $73.2_{\pm 2.0}$ | $75.5_{\pm 14.8}$ | $63.6_{\pm 10.5}$ | $70.8$ |
| 3D MedVAE | 64 | 1 | $\mathbf{83.7}_{\pm 2.8}$ | $\mathbf{87.0}_{\pm 7.3}$ | $68.4_{\pm 2.4}$ | $\mathbf{79.7}$ |
| Bicubic | 512 | 1 | $\mathbf{72.3}_{\pm 2.2}$ | $38.4_{\pm 24.5}$ | $\mathbf{59.4}_{\pm 2.5}$ | $56.7$ |
| KL-VAE | 512 | 4 | $67.7_{\pm 3.9}$ | $42.6_{\pm 4.0}$ | $50.9_{\pm 5.1}$ | $53.7$ |
| VQ-GAN | 512 | 4 | $68.9_{\pm 7.0}$ | $30.6_{\pm 12.5}$ | $57.4_{\pm 5.0}$ | $52.3$ |
| 3D MedVAE | 512 | 1 | $72.0_{\pm 3.8}$ | $\mathbf{49.1}_{\pm 19.8}$ | $58.2_{\pm 1.7}$ | $\mathbf{59.8}$ |

Table 1: **Evaluating latent representation quality with CAD tasks.** We evaluate 2D MedVAE on five 2D CAD tasks (***Top***) and 3D MedVAE on three 3D CAD tasks (***Bottom***). We report the mean AUROC and standard deviation across three random seeds. Methods that perfectly preserve clinically-relevant features (i.e. performance equals or exceeds performance when training with high-resolution images) are in **blue**.

training stage involves lifting the 2D autoencoder architecture to 3D and performing continued fine-tuning with 3D images. In total, the MedVAE family includes 4 2D autoencoders and 2 3D autoencoders trained with various downsizing factors $f$ and latent channels $C$. Extended methods and implementation details are provided in Appendix A.

## 4. Results

In order to evaluate MedVAE (Fig. 1c), we assess (1) whether downsized latent representations can effectively replace high-resolution images in CAD pipelines while maintaining per-

formance (Section 4.1); (2) whether latent representations can reduce storage requirements and improve downstream efficiency (Section 4.2); and (3) whether decoded reconstructions effectively preserve features necessary for radiologist interpretation (Section 4.3). Extended results and analysis are provided in Appendix B.

### 4.1. Latent representation quality

We first evaluate whether clinically-relevant features are preserved in MedVAE latent representations. To this end, we measure the extent to which latent representations can serve as drop-in replacements for high-resolution input images in CAD pipelines *without* any customization or modifications to CAD model architectures.

We evaluate latent representation quality using the following 8 CAD tasks: malignancy detection on 2D FFDMs (Cai et al., 2023), calcification detection on 2D FFDMs (Cai et al., 2023), BI-RADS prediction on 2D FFDMs (Nguyen et al., 2022), bone age prediction on 2D X-rays (Halabi et al., 2019), fracture detection on 2D wrist X-rays (Nagy et al., 2022), fracture detection on 3D spine CTs (Löffler et al., 2020), fracture classification on 3D head CTs (Chilamkurthy et al., 2018), and anterior cruciate liagment (ACL) and meniscal tear detection on 3D sagittal knee MRIs (Bien et al., 2018). In order to perform each of these CAD tasks, a model must rely on fine-grained, clinically-relevant features.

For each CAD task, we train a classifier (HRNet (Wang et al., 2020) in 2D settings and SEResNet (Hu et al., 2018) in 3D settings) on a training set consisting of latent representations. We then measure the difference in classification performance between models trained directly on latent representations and models trained using original, high-resolution images; this serves as an indicator of latent representation quality (e.g. a small performance difference indicates that the downsizing approach preserves diagnostic features). We compute AUROC for binary tasks and macro AUROC for multi-class tasks. We train each classifier with three random seeds, and we report results as mean AUROC $\pm$ standard deviation.

We compare MedVAE with two categories of image downsizing methods: (1) interpolation methods (nearest, bilinear, and bicubic), which are the de-facto gold standard for medical image downsizing as demonstrated by the quantity of prior work leveraging this approach (Wantlin et al., 2023; Varma et al., 2019; Zhang et al., 2022a; Huang et al., 2021), and (2) recently-introduced large-scale natural image autoencoders (KL-VAE and VQ-GAN) (Rombach et al., 2022). Due to the fact that prior work on developing large-scale 3D autoencoders has been limited, we compare our 3D MedVAE models with 2D methods by stitching 2D latent representations together across slices such that the size of the 2D latent representation matches those generated by 3D models.

We provide results for 2D and 3D CAD tasks in Table 1. Our results demonstrate that the MedVAE training approach yields high-quality latent representations for both 2D and 3D images. At a downsizing factor of $f = 16$, 2D MedVAE perfectly preserves clinically-relevant features on four out of five 2D classification tasks. Similarly, at a downsizing factor of $f = 64$, 3D MedVAE perfectly preserves relevant clinical information on two out of three 3D classification tasks (spine and skull CT fracture detection). In these cases, performance equals or exceeds performance when training with original, high-resolution images. We also observe that MedVAE consistently outperforms the natural image autoencoders KL-VAE and VQ-GAN on all classification tasks, including the two musculoskeletal tasks (bone age

prediction and wrist fracture detection) despite the fact that no musculoskeletal radiographs are used during MedVAE training; this suggests effective generalization capabilities. Our findings also show that 3D training of autoencoders leads to high-quality latent representations due to preservation of volumetric information (e.g. fractures spanning multiple slices), particularly at $f = 64$. In summary, we demonstrate that our MedVAE training procedure yields downsized latent representations that can be used as drop-in replacements for high-resolution input images in CAD pipelines.

In Appendix Tables 8 and 9, we provide ablations demonstrating the utility of our proposed two-stage training approach on latent representation quality.

### 4.2. Storage and efficiency benefits of latent representations

Next, we evaluate the extent to which downsized MedVAE latent representations can reduce storage requirements and improve downstream efficiency of CAD pipelines. Using a 2D high-resolution network and 3D squeeze-excitation network as our base CAD architectures, we report latency, throughput, and maximum batch size. Latency is the time (in milliseconds) to perform a forward pass of the network on one batch. Throughput is the number of samples that can be evaluated by the network in one second. Finally, we report the maximum batch size (in powers of 2) for a forward pass that will fit on a single A100 GPU (2D) and an A6000 GPU (3D). We assume a high-resolution image size of $1024 \times 1024$ with 1 channel for 2D settings and a volume size of $256 \times 256 \times 256$ with 1 channel for 3D settings.

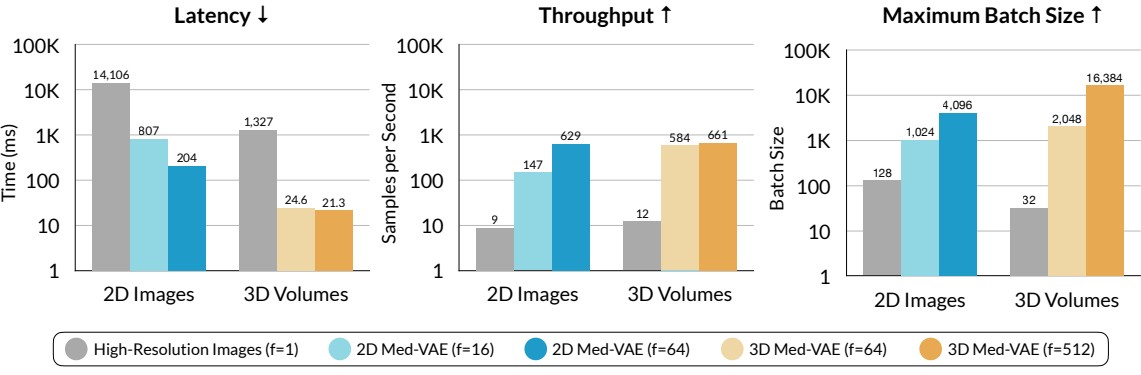

Figure 2: **CAD model efficiency.** We compare the efficiency of CAD models trained with downsized latent representations to CAD models trained with high-resolution images.

Results are provided in Figure 2. We demonstrate that training CAD models directly on downsized latent representations can lead to large improvements in model efficiency. In the 2D setting, we observe that as the downsizing factor increases to $f = 64$, latency decreases by 69x, throughput increases by 70x, and the maximum batch size increases by 32x. In the 3D setting, as the downsizing factor increases to $f = 512$, latency decreases by 62x, throughput increases by 55x, and the maximum batch size increases by 512x. Storage costs decrease proportionally with the downsizing factor (i.e. 64x for 2D and 512x for 3D).

| Method | $f$ | $C$ | FFDMs (2D) | | MSK X-rays (2D) | | Brain MRIs (3D) | | Abdomen CTs (3D) | |
|---|---|---|---|---|---|---|---|---|---|---|
| | | | PSNR ↑ | MS-SSIM ↑ | PSNR ↑ | MS-SSIM ↑ | PSNR ↑ | MS-SSIM ↑ | PSNR ↑ | MS-SSIM ↑ |
| Bicubic | 16 | 1 | 31.69 | 0.961 | 30.18 | 0.974 | 29.27 | 0.975 | 33.81 | 0.989 |
| KL-VAE | 16 | 3 | 36.11 | 0.989 | 38.29 | 0.992 | 33.23 | **0.994** | 43.51 | 0.998 |
| VQ-GAN | 16 | 3 | 35.55 | 0.986 | 36.41 | 0.990 | 32.72 | 0.992 | 40.85 | 0.997 |
| 2D MedVAE | 16 | 1 | 32.34 | 0.969 | 33.97 | 0.973 | 29.48 | 0.980 | 33.45 | 0.983 |
| 2D MedVAE | 16 | 3 | **37.57** | **0.993** | **39.41** | **0.994** | **33.99** | **0.994** | **44.95** | **0.999** |
| 3D MedVAE | 64 | 1 | – | – | – | – | 29.52 | 0.983 | 36.61 | 0.993 |

Table 2: **Evaluating reconstruction quality.** We evaluate reconstruction quality using perceptual metrics. Here, $f$ represents the downsizing factor applied to the 2D area or 3D volume of the input image and $C$ represents the number of latent channels.

### 4.3. Reconstructed image quality

We evaluate whether clinically-relevant features are preserved in reconstructed images using both automated and manual perceptual quality evaluations. These evaluations quantify the extent to which the encoding and subsequent decoding processes retain relevant features.

For automated evaluations, we use perceptual metrics to compare reconstructed images with the original inputs. We report peak signal-to-noise ratio (PSNR) and the multi-scale structural similarity index measure (MS-SSIM). For 2D evaluations, we measure perceptual quality on X-rays (Feng et al., 2021; Johnson et al., 2019); FFDMs (Jeong et al., 2022; Sorkhei et al., 2021; RSNA, 2023; Nguyen et al., 2022; Moreira et al., 2012; Cai et al., 2023); and musculoskeletal X-rays (Nagy et al., 2022). For 3D evaluations, we compute metrics on brain MRIs (Jack Jr et al., 2008; Dagley et al., 2017; Insel et al., 2020; LaMontagne et al., 2019); head CTs (Chilamkurthy et al., 2018); abdomen CTs (Ji et al., 2022); CTs from a wide range of anatomies (Wasserthal et al., 2023); lung CTs (Armato III et al., 2011); and knee MRIs (Bien et al., 2018). Results are in Table 2 and Appendix Tables 5 and 6.

We find that 2D MedVAE achieves the highest perceptual quality across all evaluated image types. In particular, our evaluations with musculoskeletal X-rays, brain MRIs, and abdomen CTs explore generalization of 2D MedVAE to unseen anatomical features; notably, 2D MedVAE achieves the highest scores on these task, despite the fact that 2D MedVAE was not trained on musculoskeletal X-rays, MRI, or CT slices. We also note a general trend that increasing the number of latent channels $C$ improves perceptual quality of the reconstructed image. We also observe that 3D MedVAE achieves competitive performance, despite utilizing a significantly higher downsizing factor than comparable 2D methods (i.e. downsizing across all three dimensions rather than just two).

We supplement our automated evaluations of reconstructed image quality with a manual reader study. Three radiologists are each presented with 50 pairs of chest X-rays containing fractures (Feng et al., 2021). Each pair consists of an original high-resolution image on the left and a reconstructed image on the right. The reconstructed images are scored on a 5-point Likert scale ranging from -2 to 2 based on three main criteria: image fidelity, preservation of diagnostic features, and the presence of artifacts. Readers rated image fidelity for 2D MedVAE to be 2.8 points higher than bicubic interpolation averaged across the two downsizing factors. 2D MedVAE also better preserved clinically-relevant features (2.8 points). Artifacts (e.g. blurring, hallucinations) were more frequent in interpolated

images (2.6 points), which severely suffered from blurring artifacts. In summary, our reader study suggests that 2D MedVAE better preserves diagnostic features than interpolation. In Figure 4, we provide qualitative examples of a reconstructed chest X-ray and a reconstructed T1-weighted brain MRI slice.

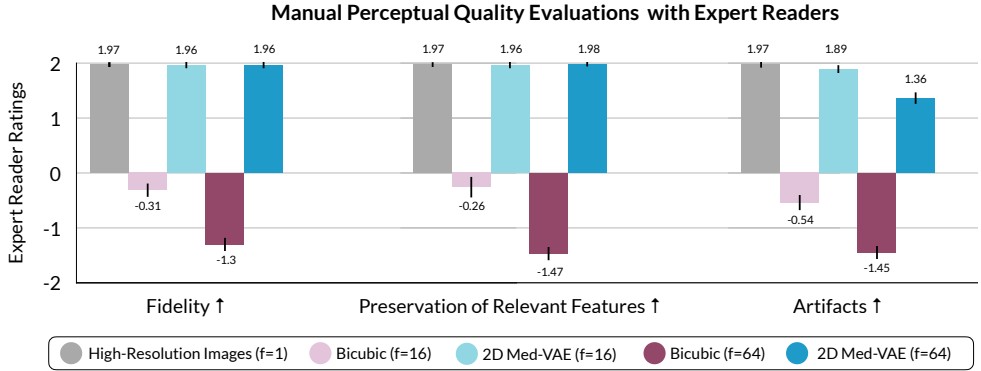

Figure 3: **Reader evaluations.** We report scores from three expert readers on fidelity, preservation of relevant features, and artifacts. Bars represent 95% confidence intervals.

## 5. Discussion

In this work, we introduced MedVAE, a family of 6 large-scale autoencoders developed using a novel two-stage training procedure. We demonstrate with extensive evaluations that (1) downsized latent representations can effectively replace high-resolution images in CAD pipelines while maintaining or exceeding performance, (2) downsized latent representations reduce storage requirements (up to 512x) and improve downstream efficiency (up to 70x in model throughput) when compared to high-resolution input images, and (3) reconstructed images effectively preserve relevant features necessary for clinical interpretation by radiologists. Our work demonstrates the potential that large-scale, generalizable autoencoders hold in addressing critical storage and efficiency challenges in the medical domain.

## Acknowledgments

MV is supported by graduate fellowship awards from the Department of Defense (ND-SEG), the Knight-Hennessy Scholars program at Stanford University, and the Quad program. AK is supported by graduate fellowships from the Knight-Hennessy Scholars program at Stanford University and Tau Beta Pi society. RS was supported by the Rubicon fellowship of the Dutch National Research Council (NWO). AC is supported by NIH grants R01 HL167974, R01HL169345, R01 AR077604, R01 EB002524, R01 AR079431, P41 EB027060, AY2 AX000045, and 1AYS AX0000024-01; ARPA-H grants AY2AX000045 and 1AYSAX0000024-01; and NIH contracts 75N92020C00008 and 75N92020C00021. A.C. has provided consulting services to Patient Square Capital, Chondrometrics GmbH, and Elucid Bioimaging; is co-founder of Cognita; has equity interest in Cognita, Subtle Medical, LVIS Corp, Brain Key. CL is supported by NIH grants R01 HL155410, R01 HL157235, by AHRQ grant R18HS026886, and by the Gordon and Betty Moore Foundation. CL is also supported by the Medical Imaging and Data Resource Center (MIDRC), which is funded by the National Institute of Biomedical Imaging and Bioengineering (NIBIB) under contract 75N92020C00021 and through the Advanced Research Projects Agency for Health (ARPA-H).

This research was funded, in part, by the Advanced Research Projects Agency for Health (ARPA-H). The views and conclusions contained in this document are those of the authors and should not be interpreted as representing the official policies, either expressed or implied, of the U.S. Government.

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

# Contents

## Appendix A. Extended Methods

### A.1. Background

In this section, we provide background information on autoencoders.

2D autoencoding methods can be formulated as follows. We begin with a training dataset $\mathcal{D} = \{x_i\}_{i=1}^N$ consisting of $N$ high-resolution input images $x_i \in \mathcal{X}$. Each high-resolution image $x_i$ has dimensions $H \times W$ with $B$ channels, which can be expressed as $x_i \in \mathbb{R}^{H \times W \times B}$. An autoencoding method learns an encoding function $g : \mathcal{X} \to \mathcal{Z}$, where $\mathcal{Z}$ represents a low-dimensional latent space and $z_i \in \mathcal{Z}$ represents the downsized latent representation corresponding to the input $x_i$. Let $f$ represent the downsizing factor applied to the 2D area of the image; then, the latent representation $z_i$ can be expressed as $z_i \in \mathbb{R}^{(H/(\sqrt{f}) \times (W/\sqrt{f}) \times C}$, where $C$ is a pre-specified number of latent channels. Autoencoding methods also learn a decoding function $h : \mathcal{Z} \to \hat{\mathcal{X}}$, which reconstructs the image $\hat{x}_i$ from the latent representation $z_i$. The encoding and decoding functions $g$ and $h$ are optimized in an end-to-end manner with the goal of maximizing perceptual similarity between $x_i$ and $\hat{x}_i$.

3D autoencoding methods follow a similar formulation, where each image $x_i$ represents a 3D volume with dimensions $H \times W \times S$ with $B$ channels. Here, the downsizing factor $f$ is applied to the 3D volume of the image; as a result, the latent representation $z_i$ can be expressed as $z_i \in \mathbb{R}^{(H/(\sqrt[3]{f}) \times (W/\sqrt[3]{f}) \times (S/(\sqrt[3]{f}) \times C}$, where $C$ is a pre-specified number of latent channels.

### A.2. Curating a large-scale training dataset

We first collect a large-scale, open-source training dataset $\mathcal{D}$ for training medical image autoencoders. We incorporate diverse modalities and anatomical features in order to ensure that trained autoencoders gain proficiency in processing the wide variety of diagnostic features that occur in medical images. Our dataset consists of 1,021,356 2D images and 31,374 3D images obtained from 19 multi-institutional, open-source datasets.

2D images include chest X-rays and FFDMs, selected because (a) chest X-rays are well-studied with large amounts of publicly-available data and (b) FFDMs are a challenging class of images due to large dimensions and the presence of fine-grained features critical

for diagnoses (e.g. microcalcifications). We collect images from two chest X-ray datasets and six FFDM datasets (Johnson et al., 2019; Feng et al., 2021; Jeong et al., 2022; Sorkhei et al., 2021; RSNA, 2023; Nguyen et al., 2022; Moreira et al., 2012; Cai et al., 2023).

3D images include head MRIs, knee MRIs, and high-resolution whole-body (head, neck, abdomen, chest, lower limb) CTs. We selected these datasets since (a) head MRIs/CTs are a commonly obtained examination, and (b) high-resolution CTs tend to contain subtle features and consume large amounts of storage. These images were curated from four T1- and T2-weighted head MRI datasets (14,296), one knee MRI dataset (3,564), two head/neck CT datasets (10,156), two whole-body CT datasets (1,434), and two chest CT datasets (1,924) (Jack Jr et al., 2008; Dagley et al., 2017; Insel et al., 2020; LaMontagne et al., 2019; Bien et al., 2018; Hooper et al., 2021; Chilamkurthy et al., 2018; Wasserthal et al., 2023; Ji et al., 2022; Armato III et al., 2011; Stanford Center for Artificial Intelligence in Medicine & Imaging (AIMI), 2024).

### A.3. Training autoencoders for medical images

In this section, we discuss our two-stage approach for training generalizable autoencoders for medical images. Motivated by prior work on natural images (Rombach et al., 2022), we elect to use variational autoencoders (VAEs) as our backbone. In the first stage of training, we optimize for reconstruction quality by maximizing perceptual similarity between the input image $x$ and the reconstructed image $\hat{x}$. Whereas existing works train autoencoders solely using this approach, the medical image domain introduces the added complexity of subtle, fine-grained features required for clinical interpretation of images; thus, we introduce a second stage of training, where the latent representation space $\mathcal{Z}$ is refined with continued fine-tuning. Our approach is intended to explicitly preserve diverse clinically-relevant features in both latent representations and reconstructed images. In total, the Med-VAE family includes four 2D VAEs and two 3D VAEs trained with various downsizing factors.

**Stage 1: Training Base Autoencoders** (Fig. 1a). We begin by performing base training of the autoencoders using the collected 2D images in order to optimize the quality of reconstructions $\hat{x}$. In line with prior work (Rombach et al., 2022), each Med-VAE autoencoder learns an encoder and decoder (corresponding to functions $g$ and $h$) end-to-end using a fully convolutional VAE. Each Med-VAE autoencoder accepts single-channel, high-resolution medical images $x_i$ as input, applies function $g$ to transform the input to a downsized latent representation $z_i$, and then applies function $h$ to reconstruct the original image $\hat{x}_i$. Med-VAE models are characterized by two hyperparameters: $f$, which represents the downsizing factor applied to the 2D area of the input image, and $C$, which describes the number of channels included in the latent representation. For instance, given an input image $x_i$ of size $H \times W \times 1$, a Med-VAE model with $f = 16$ and $C = 3$ would generate a latent representation $z_i$ of size $(H/4) \times (W/4) \times 3$, downsizing the image area by 16x and adding two additional channels. The reconstructed image $\hat{x}_i$ would be of size $H \times W \times 1$.

In order to learn functions $g$ and $h$, the VAE is trained to maximize the similarity between $x_i$ and $\hat{x}_i$ using a perceptual loss term (Zhang et al., 2018) and a patch-based adversarial objective (Isola et al., 2018). Additionally, in order to ensure preservation of clinically-relevant features within the reconstructed image, we introduce a domain-specific

embedding consistency loss based on BiomedCLIP, a pretrained vision-language foundation model trained on a large corpus of paired medical image-text data (Zhang et al., 2023). During training, we apply an $L_2$ penalty between BiomedCLIP embeddings corresponding to the input image $x_i$ and the reconstructed image $\hat{x}_i$. This loss function is inspired by prior work on developing autoencoders for chest X-rays (Lee et al., 2023). Finally, in addition to the loss functions listed above, a KL-divergence penalty is applied to the latent sample in order to pull latents towards a standard normal; the penalty is assigned a low weight of 1e-6.

We use the above loss functions and the curated dataset of one million 2D images to train the following four base autoencoders, trained across various downsizing factors and latent channels. Implementation details for each base model is described below:

- **2D Base Autoencoder (Stage 1) with $f = 16$ and $C = 1$**: This autoencoder yields latent representations $z_i$ of size $(H/4) \times (W/4) \times 1$. Stage 1 training is performed from scratch. The VAE is trained solely with the perceptual loss, the KL-divergence penalty, and the BiomedCLIP embedding consistency loss for the first 3125 steps; then, the patch-based adversarial objective is applied. We train for 100K steps using 8 NVIDIA A100 GPUs and a batch size of 32.

- **2D Base Autoencoder (Stage 1) with $f = 16$ and $C = 3$**: This autoencoder yields latent representations $z_i$ of size $(H/4) \times (W/4) \times 3$. We first initialize the VAE with weights from a previously-developed natural image autoencoder (KL-VAE) (Rombach et al., 2022). Then, we perform Stage 1 training using LoRA (Hu et al., 2021) with rank=4 applied to all 2D convolutional layers. We train with all four loss functions for 50k steps using 8 A100 GPUs and a batch size of 32.

- **2D Base Autoencoder (Stage 1) with $f = 64$ and $C = 1$**: This autoencoder yields latent representations $z_i$ of size $(H/8) \times (W/8) \times 1$. Stage 1 training is performed from scratch. The VAE is trained solely with the perceptual loss, the KL-divergence penalty, and the BiomedCLIP embedding consistency loss for the first 3125 steps; then, the patch-based adversarial objective is applied. We train for 100K steps using 8 NVIDIA A100 GPUs and a batch size of 32.

- **2D Base Autoencoder (Stage 1) with $f = 64$ and $C = 4$**: This autoencoder yields latent representations $z_i$ of size $(H/8) \times (W/8) \times 4$. We first initialize the VAE with weights from a previously-developed natural image autoencoder (KL-VAE) (Rombach et al., 2022). Then, we perform Stage 1 training using LoRA (Hu et al., 2021) with rank=4 applied to all 2D convolutional layers. We train with all four loss functions for 50k steps using 8 A100 GPUs and a batch size of 32.

**Stage 2: Preserving Clinically-Relevant Features Across Modalities** (Fig. 1b). After performing base training of the autoencoders using the collected 2D images, we introduce a second stage of training intended to further refine the latent space such that clinically-relevant features are preserved across various modalities.

In the context of 2D imaging modalities, the second training stage takes the form of a lightweight fine-tuning procedure designed to maximize consistency in clinically-relevant features between the input image and the latent representation. Our key insight here is that

image embeddings generated by BiomedCLIP (Zhang et al., 2023) can effectively capture clinically-relevant features in 2D medical images, suggesting utility as a guidance mechanism during training[2]. We freeze all parameters in the encoder and decoder of the VAE. During training, the input image $x_i$ is passed through the frozen VAE encoder to generate the latent representation $z_i$; then, $z_i$ is passed through a series of lightweight, trainable projection layers, which yield an output representation $\bar{z}_i$ with the same size as $z_i$. Let the function $b(\cdot)$ represent the BiomedCLIP embedding function. We optimize the projection layer weights using a domain-specific embedding consistency loss, which takes the form of an $L_2$ loss between $b(x_i)$ and $b(\bar{z}_i)$. All downstream evaluations of latent representation quality are performed with the projected latent $\bar{z}_i$. We perform Stage 2 training using the curated 2D training dataset with one million images. Our procedure yields four 2D Med-VAE autoencoders with various downsizing factors and number of latent channels:

- **2D Med-VAE with** $f = 16$ **and** $C = 1$: The projection layers generate $\bar{z}_i$ of size $(H/4) \times (W/4) \times 1$. Stage 2 training is performed for 50K steps using 8 NVIDIA A100 GPUs and a batch size of 32.

- **2D Med-VAE with** $f = 16$ **and** $C = 3$: The projection layers generate $\bar{z}_i$ of size $(H/4) \times (W/4) \times 3$. Stage 2 training is performed for 50K steps using 8 NVIDIA A100 GPUs and a batch size of 32.

- **2D Med-VAE with** $f = 64$ **and** $C = 1$: The projection layers generate $\bar{z}_i$ of size $(H/8) \times (W/8) \times 1$. Stage 2 training is performed for 60K steps using 8 NVIDIA A100 GPUs and a batch size of 32.

- **2D Med-VAE with** $f = 64$ **and** $C = 4$: The projection layers generate $\bar{z}_i$ of size $(H/8) \times (W/8) \times 4$. Stage 2 training is performed for 50K steps using 8 NVIDIA A100 GPUs and a batch size of 32.

In the context of 3D imaging modalities (e.g. CT scans, MRIs), the second training stage involves lifting the 2D VAE architecture to 3D using a kernel centering inflation strategy (Zhang et al., 2022b); we then continue training with 3D images. We note here that using external 2D medical foundation models like BiomedCLIP to enforce feature consistency is inadequate for 3D settings. As a result, we instead implement a training procedure focused on maximizing perceptual similarity, analogous to 2D stage 1 training. We train the 3D autoencoders using random cubic patches of size $64 \times 64 \times 64$. The perceptual loss and the patch-based adversarial objective are calculated per-slice, with the final loss term computed as the mean across all slices in the volume. Following such a training strategy, a 3D Med-VAE model with $f = 64$, $C = 1$, and input image $x_i$ of size $H \times W \times S \times 1$ would generate a latent representation $z_i$ of size $(H/4) \times (W/4) \times (S/4) \times 1$, downsizing the volume by 64x. We perform Stage 2 training using the curated dataset of 31,374 3D images. Our procedure yields two 3D Med-VAE autoencoders across various downsizing factors:

- **3D Med-VAE with** $f = 64$ **and** $C = 1$: The latent representations $z_i$ are of size $(H/4) \times (W/4) \times (S/4) \times 1$. We initialize the VAE with weights from 2D Base

---

2. We use the `BiomedCLIP-PubMedBERT_256-vit_base_patch16_224` model available on HuggingFace at https://huggingface.co/microsoft/BiomedCLIP-PubMedBERT_256-vit_base_patch16_224.

| Classification Task | Dimensionality | Classes | Dataset | Modality | Anatomy | Num. Images |
|---|---|---|---|---|---|---|
| Malignancy Detection | 2D | 2 | CMMD | FFDM | Breast | 3744 |
| Calcification Detection | 2D | 2 | CMMD | FFDM | Breast | 5202 |
| BI-RADS Classification | 2D | 5 | VinDR-Mammo | FFDM | Breast | 20,000 |
| Bone Age Prediction | 2D | 20 | RSNA Bone Age | X-Ray | Hand | 14,036 |
| Wrist Fracture Detection | 2D | 2 | GRAZPEDWRI-DX | X-Ray | Wrist | 14,113 |
| Spine Fracture Detection | 3D | 2 | VerSe | CT | Spine | 160 |
| Head Fracture Detection | 3D | 2 | CQ500 | CT | Head | 378 |
| ACL/Meniscal Tear Detection | 3D | 2 | MRNet | MRI | Knee | 1250 |

Table 3: **Summary of CAD tasks used for evaluating latent representation quality**. We report the task name, number of classes associated with the task, the dataset name, imaging modality, anatomical features, and the number of images after preprocessing (Cai et al., 2023; Nguyen et al., 2022; Halabi et al., 2019; Nagy et al., 2022; Löffler et al., 2020; Chilamkurthy et al., 2018; Bien et al., 2018).

Autoencoder (Stage 1) with $f = 16$ and $C = 1$. We then train the VAE for 35K steps using 4 NVIDIA A6000 GPUs and a batch size of 32.

- **3D Med-VAE with $f = 512$ and $C = 1$**: The latent representations $z_i$ are of size $(H/8) \times (W/8) \times (S/8) \times 1$. We initialize the VAE with weights from 2D Base Autoencoder (Stage 1) with $f = 64$ and $C = 1$. We then train the VAE for 140K steps using 1 NVIDIA A6000 GPU and a batch size of 8. Both 3D Med-VAEs are trained for the same number of steps when accounting for batch size.

## Appendix B. Extended Results

### B.1. Evaluating latent representations

We evaluate the quality of latent representations $z$ with a set of eight clinically-relevant CAD tasks, which directly evaluate the preservation of clinically-relevant features in 2D and 3D images (Table 3). For each CAD task, we measure the difference in classification performance between models trained using latent representations and those trained using original, high-resolution images; this serves as an indicator of latent quality by directly measuring the retention of important diagnostic features. These evaluations also provide insights into potential performance gains afforded by training downstream models directly on Med-VAE latent representations rather than high-resolution images.

Below, we provide implementation details for each 2D CAD task.

1. **Malignancy Detection:** We evaluate the quality of FFDM latent representations on a binary malignancy detection task, which involves predicting the presence or absence of a malignancy. We use images from the Chinese Mammography Dataset (CMMD), which includes a total of 5202 deidentified FFDMs from 1775 patients (Cai et al., 2023; Cui et al., 2021). CMMD includes labels indicating the presence of masses and calcifications as well as biopsy-confirmed labels indicating benign and malignant findings. We assigned 80% of patients to the training set (1420 patients with 2982

images) and the remaining 20% to the test set (355 patients with 762 images). The average size of an FFDM after preprocessing was $1999.2 \times 793.9 \times 1$. In order to maintain consistent sizing, we downsized each FFDM to $1024 \times 512 \times 1$ using bicubic interpolation.

2. **Calcification Detection:** We evaluate the quality of FFDM latent representations on a binary calcification detection task, which involves identifying the presence or absence of breast calcifications. We use the CMMD dataset, described in detail above (Cui et al., 2021; Cai et al., 2023). We preprocessed the CMMD dataset by assigning 80% of patients to the training set (1420 patients with 4156 images) and 20% of patients to the test set (355 patients with 1046 images).

3. **BI-RADS Classification:** We evaluate the quality of FFDM latent representations on Breast Imaging Reporting and Data System (BI-RADS) classification. We use images from the VinDR-Mammo dataset, which includes a total of 20,000 deidentified FFDMs from 5000 studies collected from Hanoi Medical University Hospital and Hospital 108 in Vietnam (Nguyen et al., 2022). BI-RADS scores evaluate the likelihood of cancer on an integer scale from 0 to 6(Nguyen et al., 2022). We use the provided data splits for VinDR-Mammo, which assign 16,000 images to the training set and 4000 images to the test set. There are no images with BI-RADS scores of 0 or 6. The average size of an FFDM after preprocessing was $2607.3 \times 948.6 \times 1$. In order to maintain consistent sizing across the dataset, we downsized each X-ray to $1024 \times 512 \times 1$.

4. **Bone Age Prediction:** We evaluate the quality of musculoskeletal X-ray latent representations on a bone age prediction task. We use images from the RSNA Bone Age dataset, which includes 14,036 hand radiographs collected from Children's Hospital Colorado and Lucile Packard Children's Hospital at Stanford University (Halabi et al., 2019). We use the provided data splits for the RSNA Bone Age dataset, which assign 12,611 images to the training set and 1425 images to the test set. The average size of a musculoskeletal X-ray after preprocessing was $1665.4 \times 1319.8 \times 1$. In order to maintain consistent sizing across the dataset, we downsized each X-ray to $1024 \times 1024 \times 1$.

5. **Pediatric Wrist Fracture Detection:** We evaluate the quality of musculoskeletal X-ray latent representations on a binary wrist fracture detection task. We use images from the GRAZPEDWRI-DX dataset, which includes a total of 20,327 deidentified images from 6,091 patients collected at University Hospital Graz in Austria (Nagy et al., 2022). We preprocessed the GRAZPEDWRI-DX dataset by first using provided labels to remove all samples with metal hardware and casts, which may exhibit spurious correlations with the target labels. We then assigned 75% of patients to the training set (4281 patients with 10,511 images) and the remaining 25% to the test set (1428 patients with 3602 images). The average size of a musculoskeletal X-ray after preprocessing was $987.8 \times 537.7 \times 1$. In order to maintain consistent sizing across the dataset, we resized each X-ray to $1024 \times 512 \times 1$.

We perform each 2D CAD task listed above using a pretrained HRNet_w64 neural network implemented in the `timm` Python package(Wang et al., 2020; Wightman, 2019). HR-

Nets are a type of convolutional neural network adapted for classification of high-resolution images. We preprocess latent representations by applying the mean operation across the channel dimension if more than one channel is present. We train the HRNet on 2 A100 GPUs using supervised linear probing with one output class. We train for 100 epochs using a batch size of 256, an AdamW optimizer (Loshchilov and Hutter, 2019) with an initial learning rate of 1e-4, and cross-entropy loss. Classification performance is measured on the test set using the final model checkpoint. We report AUROC for binary classification tasks and Macro AUROC for multi-class classification tasks.

Below, we provide implementation details for each 3D CAD task.

1. **Spine Fracture Detection:** We evaluate the quality of Spine CT latent representations on a binary spine fracture detection task. We use images from the VerSe 2019 dataset (Löffler et al., 2020), which includes 160 high-resolution, 1-mm isotropic or in sagittal 2-mm to 3-mm series of 1-mm in-plane resolution, spine CT images. The training, validation, and testing split (50/25/25) was maintained from the original dataset. The final size of a volume after preprocessing was $224 \times 224 \times 160$.

2. **Head Fracture Detection:** We evaluate the quality of head CT latent representations on a binary head fracture detection task. We use images from the CQ500 dataset (Chilamkurthy et al., 2018), which includes 378 head CT images. This dataset was curated by the Centre for Advanced Research in Imaging, Neurosciences, and Genomics (CARING) in New Delhi, India. Images were divided into training and testing sets following an 80/20 split. The final size of a volume after preprocessing was $224 \times 224 \times 44$.

3. **ACL and Meniscal Tear Detection:** We evaluate the quality of knee MRI latent representations on a binary ACL or meniscal tear detection task. We use images from the MRNet dataset (Bien et al., 2018), which includes 1250 sagittal knee MRI scans performed at Stanford University Medical Center between 2001-2012. A positive label in this context may indicate the presence of an ACL tear, a meniscal tear, or both simultaneously. The dataset was split into a training and test set (95/5). The final size of a volume after preprocessing was $56 \times 256 \times 256$.

We perform each 3D CAD task listed above using the MONAI SEResNet-152 (Hu et al., 2018) architecture. We implemented a weighted sampling strategy for the head fracture detection and ACL and meniscal tear detection tasks due to class imbalance. We trained the SEResNet-152 on an A6000 GPU using supervised linear probing with 1 output class. We trained for 100 epochs with a batch size of 20 for latents, a batch size of 10 for the original images, an AdamW optimizer (Loshchilov and Hutter, 2019) with an initial learning rate of 1e-4, and binary cross-entropy loss. Classification performance (AUROC) is measured on the test set using the final model checkpoint.

For latent representation evaluations, we report classification performance using AUROC, calculated using the `torchmetrics` library. We report mean and standard deviations across three runs with different random seeds.

In Table 4, we compare performance of 2D MedVAE and 3D MedVAE on 3D CAD tasks. These findings demonstrate that 3D training of autoencoders leads to high-quality

| Method | $f$ | $C$ | AUROC ↑ | | | |
|---|---|---|---|---|---|---|
| | | | Spine Fractures | Skull Fractures | Knee Injury | Average |
| High-Resolution | 1 | 1 | $\mathbf{82.9}_{\pm 2.2}$ | $\mathbf{63.9}_{\pm 6.3}$ | $\mathbf{69.9}_{\pm 0.6}$ | $\mathbf{72.2}$ |
| 2D Med-VAE | 64 | 1 | $80.5_{\pm 4.9}$ | $57.4_{\pm 4.0}$ | $67.3_{\pm 3.6}$ | $68.4$ |
| 2D Med-VAE | 64 | 3 | $78.6_{\pm 0.8}$ | $50.9_{\pm 19.5}$ | $60.9_{\pm 4.2}$ | $63.5$ |
| 3D Med-VAE | 64 | 1 | $\mathbf{83.7}_{\pm 2.8}$ | $\mathbf{87.0}_{\pm 7.3}$ | $68.4_{\pm 2.4}$ | $\mathbf{79.7}$ |
| 2D Med-VAE | 512 | 1 | $65.9_{\pm 8.7}$ | $\mathbf{63.0}_{\pm 1.1}$ | $55.9_{\pm 8.3}$ | $\mathbf{61.6}$ |
| 2D Med-VAE | 512 | 4 | $\mathbf{81.9}_{\pm 1.2}$ | $17.1_{\pm 8.6}$ | $52.6_{\pm 1.9}$ | $50.5$ |
| 3D Med-VAE | 512 | 1 | $72.0_{\pm 3.8}$ | $49.1_{\pm 19.8}$ | $\mathbf{58.2}_{\pm 1.7}$ | $59.8$ |

Table 4: **Comparing 2D Med-VAE and 3D Med-VAE on 3D CAD tasks.** We compare 3D Med-VAE with 2D Med-VAE models. For 2D Med-VAE, we stitch 2D latent representations together across slices such that the size of the 2D latent representation matches those generated by the 3D model. Here, $f$ represents the downsizing factor applied to the 3D volume of the input image and $C$ represents the number of latent channels. The best performing models on each task are bolded. We highlight methods that perfectly preserve clinically-relevant features in blue.

latent representations due to preservation of volumetric information (e.g. fractures spanning multiple slices), particularly at f = 64.

## B.2. Evaluating reconstructed images

We evaluate the quality of reconstructions $\hat{x}$ using both automated and manual perceptual quality evaluations. Perceptual quality assessments measure information loss resulting from the autoencoding process by comparing the original image to the reconstructed (decoded) image. These evaluations quantify the extent to which the encoding and subsequent decoding process retains relevant features.

For 2D images, we evaluate full-image perceptual quality on chest X-rays, FFDMs, and musculoskeletal X-rays; we also evaluate fine-grained perceptual quality on musculoskeletal X-rays. Chest X-rays are obtained from CANDID-PTX (Feng et al., 2021) and MIMIC-CXR (Johnson et al., 2019); FFDMs are obtained from RSNA Mammography (RSNA, 2023), VinDR-Mammo (Nguyen et al., 2022), CSAW-CC (Sorkhei et al., 2021), EMBED (Jeong et al., 2022), CMMD (Cai et al., 2023), and INBreast (Moreira et al., 2012); musculoskeletal X-rays are obtained from GRAZPEDWRI-DX (Nagy et al., 2022). We compute two standard perceptual quality metrics: PSNR and MS-SSIM. For 2D fine-grained perceptual quality evaluations, we extract 7677 images containing fractures from GRAZPEDWRI-DX, and we use bounding boxes provided by the authors to isolate the region of the fracture (Nagy et al., 2022). We then compute PSNR scores on these regions.

For 3D full-volume perceptual quality evaluations, we evaluate full-image perceptual quality on head MRIs, head CTs, abdomen CTs, whole-body CTs, lung CTs, and knee MRIs. Head MRIs are obtained from Alzheimer's Disease Neuroimaging Initiative (ADNI) (Jack Jr et al., 2008), Harvard Aging Brain Study (HABS) (Dagley et al., 2017), A4 dataset

| Method | $f$ | $C$ | Mammograms | | Chest X-rays | | Musculoskeletal X-rays | | Wrist X-rays (FG) |
|---|---|---|---|---|---|---|---|---|---|
| | | | PSNR ↑ | MS-SSIM ↑ | PSNR ↑ | MS-SSIM ↑ | PSNR ↑ | MS-SSIM ↑ | PSNR ↑ |
| Nearest | 16 | 1 | $25.95_{\pm0.06}$ | $0.846_{\pm0.00}$ | $29.87_{\pm0.04}$ | $0.942_{\pm0.00}$ | $24.06_{\pm0.02}$ | $0.890_{\pm0.00}$ | $26.11_{\pm0.02}$ |
| Bilinear | 16 | 1 | $30.18_{\pm0.07}$ | $0.936_{\pm0.00}$ | $34.23_{\pm0.03}$ | $0.981_{\pm0.00}$ | $28.75_{\pm0.02}$ | $0.959_{\pm0.00}$ | $30.92_{\pm0.03}$ |
| Bicubic | 16 | 1 | $31.69_{\pm0.07}$ | $0.961_{\pm0.00}$ | $35.48_{\pm0.03}$ | $0.989_{\pm0.00}$ | $30.18_{\pm0.02}$ | $0.974_{\pm0.00}$ | $32.65_{\pm0.04}$ |
| KL-VAE | 16 | 3 | $36.11_{\pm0.07}$ | $0.989_{\pm0.00}$ | $41.45_{\pm0.04}$ | $0.996_{\pm0.00}$ | $38.29_{\pm0.03}$ | $0.992_{\pm0.00}$ | $36.55_{\pm0.03}$ |
| VQ-GAN | 16 | 3 | $35.55_{\pm0.07}$ | $0.986_{\pm0.00}$ | $37.80_{\pm0.03}$ | $0.995_{\pm0.00}$ | $36.41_{\pm0.02}$ | $0.990_{\pm0.00}$ | $34.19_{\pm0.04}$ |
| 2D Med-VAE | 16 | 1 | $32.34_{\pm0.07}$ | $0.969_{\pm0.00}$ | $38.44_{\pm0.02}$ | $0.990_{\pm0.00}$ | $33.97_{\pm0.03}$ | $0.973_{\pm0.00}$ | $31.97_{\pm0.03}$ |
| 2D Med-VAE | 16 | 3 | $\mathbf{37.57}_{\pm0.08}$ | $\mathbf{0.993}_{\pm0.00}$ | $\mathbf{43.55}_{\ \pm0.02}$ | $\mathbf{0.997}_{\pm0.00}$ | $\mathbf{39.41}_{\pm0.04}$ | $\mathbf{0.994}_{\pm0.00}$ | $\mathbf{37.61}_{\pm0.02}$ |
| Nearest | 64 | 1 | $22.46_{\pm0.05}$ | $0.669_{\pm0.00}$ | $26.22_{\pm0.03}$ | $0.858_{\pm0.00}$ | $19.93_{\pm0.02}$ | $0.756_{\pm0.00}$ | $22.14_{\pm0.04}$ |
| Bilinear | 64 | 1 | $26.81_{\pm0.06}$ | $0.837_{\pm0.00}$ | $31.18_{\pm0.03}$ | $0.949_{\pm0.00}$ | $24.89_{\pm0.01}$ | $0.898_{\pm0.00}$ | $27.12_{\pm0.03}$ |
| Bicubic | 64 | 1 | $27.84_{\pm0.06}$ | $0.874_{\pm0.00}$ | $32.09_{\pm0.03}$ | $0.962_{\pm0.00}$ | $25.92_{\pm0.01}$ | $0.922_{\pm0.00}$ | $28.54_{\pm0.03}$ |
| KL-VAE | 64 | 4 | $31.88_{\pm0.07}$ | $0.959_{\pm0.00}$ | $36.37_{\pm0.01}$ | $0.987_{\pm0.00}$ | $33.49_{\pm0.02}$ | $0.966_{\pm0.00}$ | $31.04_{\pm0.03}$ |
| VQ-GAN | 64 | 4 | $30.13_{\pm0.06}$ | $0.938_{\pm0.00}$ | $34.87_{\pm0.02}$ | $0.980_{\pm0.00}$ | $32.00_{\pm0.02}$ | $0.953_{\pm0.0}$ | $29.92_{\pm0.02}$ |
| 2D Med-VAE | 64 | 1 | $28.00_{\pm0.07}$ | $0.872_{\pm0.00}$ | $31.92_{\pm0.04}$ | $0.962_{\pm0.00}$ | $28.27_{\pm0.02}$ | $0.917_{\pm0.00}$ | $28.03_{\pm0.01}$ |
| 2D Med-VAE | 64 | 4 | $\mathbf{33.13}_{\pm0.07}$ | $\mathbf{0.969}_{\pm0.00}$ | $\mathbf{38.88}_{\pm0.03}$ | $\mathbf{0.990}_{\pm0.00}$ | $\mathbf{34.73}_{\pm0.02}$ | $\mathbf{0.972}_{\pm0.00}$ | $\mathbf{32.30}_{\pm0.02}$ |

Table 5: **Evaluating reconstruction quality on 2D datasets.** We evaluate 2D Med-VAE with perceptual quality metrics on mammograms and chest X-rays, which we classify as *in-distribution*, since the Med-VAE training set includes mammograms and chest X-rays. We also evaluate Med-VAE on musculoskeletal X-rays and wrist X-rays (fine-grained), which we classify as *out-of-distribution*. Here, $f$ represents the downsizing factor applied to the 2D area of the input image and $C$ represents the number of latent channels. The best performing models are bolded. We calculate PSNR and MS-SSIM using a random sample of 1000 images for each image type; we report mean and standard deviations across four runs with different random seeds.

(Insel et al., 2020), and Open Access Series of Imaging Studies (OASIS) brain dataset (La-Montagne et al., 2019); head CTs are obtained from CQ500 (Chilamkurthy et al., 2018); whole-body CTs are obtained from TotalSegmentator dataset (Wasserthal et al., 2023); abdomen CTs are obtained from the Abdominal Multi-Organ Segmentation (AMOS) dataset (Ji et al., 2022); lung CTs are obtained from LIDC-IDRI (Armato III et al., 2011); and knee MRIs are obtained from MRNet (Bien et al., 2018). For each volume, a center crop of volume dimensions $160 \times 160 \times 160$ was extracted. For the AMOS and CQ500 datasets, the crop region was expanded to dimensions $320 \times 320 \times 160$ to include both soft-tissue and bony features. We compute two standard perceptual quality metrics: PSNR and MS-SSIM.

In Table 5 and Table 6, we provided an extended version of Table 2 with additional perceptual quality evaluations. In Table 7, we compare 3D Med-VAE with a model referred to as 2D Med-VAE-Decoder, which has a comparable downsizing factor $f$. The 2D Med-VAE-Decoder model performs downsizing on individual 2D slices, which are then stitched and interpolated together to form a latent representation of equivalent size to the 3D Med-VAE model; we then perform fine-tuning of the decoder using our curated dataset of 3D volumes. The superiority of 3D Med-VAE to the 2D Med-VAE-Decoder approach demonstrates the utility of 3D training of autoencoders, which enables the model to capture important volumetric patterns.

For manual evaluations of reconstructed image quality, we perform a reader study with 3 radiologists. Each expert reader is presented with a pair of chest X-rays, consisting of an original high-resolution image $x$ on the left and a reconstructed image $\hat{x}$ on the right (Fig. 5). A total of 50 unique chest X-rays with fractures, randomly sampled from CANDID-PTX,

| Method | $f$ | $C$ | Brain MRIs | | Head CTs | | Abdomen CTs | | TS CTs | | Lung CTs | | Knee MRIs | |
|---|---|---|---|---|---|---|---|---|---|---|---|---|---|---|
| | | | PSNR ↑ | MS-SSIM ↑ | PSNR ↑ | MS-SSIM ↑ | PSNR ↑ | MS-SSIM ↑ | PSNR ↑ | MS-SSIM ↑ | PSNR ↑ | MS-SSIM ↑ | PSNR ↑ | MS-SSIM ↑ |
| Bicubic | 16 | 1 | 29.27 | 0.975 | 36.21 | 0.996 | 33.81 | 0.989 | 27.33 | 0.972 | 28.00 | 0.973 | 26.37 | 0.986 |
| KL-VAE | 16 | 3 | 33.23 | **0.994** | 47.65 | **1.000** | 43.51 | 0.998 | 34.14 | 0.994 | 32.62 | **0.989** | 31.31 | **0.998** |
| VQ-GAN | 16 | 3 | 32.72 | 0.992 | 42.87 | 0.999 | 40.85 | 0.997 | 33.55 | 0.993 | 32.20 | **0.989** | 30.75 | 0.997 |
| 2D Med-VAE | 16 | 1 | 29.48 | 0.980 | 39.71 | 0.997 | 33.45 | 0.983 | 29.70 | 0.983 | 28.40 | 0.973 | 27.38 | 0.990 |
| 2D Med-VAE | 16 | 3 | **33.99** | **0.994** | **48.56** | **1.000** | **44.95** | **0.999** | **34.83** | **0.995** | **33.34** | **0.989** | **31.52** | 0.997 |
| 3D Med-VAE | 64 | 1 | 29.52 | 0.983 | 39.03 | 0.999 | 36.61 | 0.993 | 31.35 | 0.987 | 28.79 | 0.975 | 28.25 | 0.994 |
| Bicubic | 64 | 1 | 26.25 | 0.911 | 30.11 | 0.980 | 28.84 | 0.955 | 24.24 | 0.914 | 24.40 | 0.928 | 24.11 | 0.956 |
| KL-VAE | 64 | 3 | 29.32 | **0.977** | 40.95 | 0.997 | 38.07 | **0.995** | 29.85 | 0.982 | 28.83 | 0.974 | 27.68 | **0.993** |
| VQ-GAN | 64 | 3 | 27.43 | 0.967 | 39.02 | 0.997 | 36.25 | 0.991 | 27.47 | 0.972 | 26.66 | 0.964 | 25.95 | 0.990 |
| 2D Med-VAE | 64 | 1 | 25.66 | 0.920 | 33.10 | 0.988 | 29.51 | 0.967 | 24.50 | 0.922 | 24.39 | 0.933 | 24.48 | 0.973 |
| 2D Med-VAE | 64 | 3 | **29.34** | 0.976 | **41.98** | **0.999** | **39.49** | **0.995** | **30.35** | **0.984** | **29.59** | **0.977** | **28.05** | **0.993** |
| 3D Med-VAE | 512 | 1 | 26.23 | 0.937 | 30.85 | 0.991 | 29.47 | 0.960 | 26.34 | 0.949 | 24.76 | 0.934 | 24.36 | 0.977 |

Table 6: **Evaluating reconstruction quality on 3D datasets.** We evaluate 3D Med-VAE with perceptual quality metrics on head MRIs, head CTs, abdomen CTs, various high-resolution CTs (TS), lung CTs, and knee MRIs. $f$ represents the downsizing factor applied to the input volume and $C$ represents the number of latent channels. The best performing models are bolded. We compare 3D Med-VAE with several 2D methods, including 2D Med-VAE, KL-VAE, and VQ-GAN.

| Method | $f$ | $C$ | Brain MRIs | | Head CTs | | Abdomen CTs | | TS CTs | | Lung CTs | | Knee MRIs | |
|---|---|---|---|---|---|---|---|---|---|---|---|---|---|---|
| | | | PSNR ↑ | MS-SSIM ↑ | PSNR ↑ | MS-SSIM ↑ | PSNR ↑ | MS-SSIM ↑ | PSNR ↑ | MS-SSIM ↑ | PSNR ↑ | MS-SSIM ↑ | PSNR ↑ | MS-SSIM ↑ |
| 2D Med-VAE-Decoder | 64 | 1 | 28.88 | 0.978 | 35.01 | 0.997 | 31.47 | 0.983 | 29.96 | 0.981 | 27.54 | 0.965 | 27.04 | 0.992 |
| 3D Med-VAE | 64 | 1 | **29.52** | **0.983** | **39.03** | **0.999** | **36.61** | **0.993** | **31.35** | **0.987** | **28.79** | **0.975** | **28.25** | **0.994** |
| 2D Med-VAE-Decoder | 512 | 1 | 25.85 | 0.927 | 18.65 | 0.824 | 20.47 | 0.699 | 25.26 | 0.929 | 23.33 | 0.909 | 23.92 | 0.969 |
| 3D Med-VAE | 512 | 1 | **26.23** | **0.937** | **30.85** | **0.991** | **29.47** | **0.960** | **26.34** | **0.949** | **24.76** | **0.934** | **24.36** | **0.977** |

Table 7: **Comparisons of 3D Med-VAE and 2D Med-VAE Decoder.** The 2D Med-VAE-Decoder model performs downsizing on individual 2D slices, which are then stitched and interpolated together to form a latent representation of equivalent size to the 3D Med-VAE model; we then perform fine-tuning of the decoder using our curated dataset of 3D volumes. We compare perceptual quality of reconstructed volumes across six 3D image types. Here, $f$ represents the downsizing factor applied to the 3D volume of the input image and $C$ represents the number of latent channels. The best performing models on each task are bolded.

are selected and presented in a randomized order (Feng et al., 2021). The reader study poses three distinct questions on image fidelity, preservation of clinically-relevant features, and the presence of artifacts. Each question is scored based on a 5-point Likert scale ranging between -2 and 2. Below, we provide additional details on each of these questions:

1. **Image Fidelity:** This question aims to assess how closely the reconstructed CXR image resembles the original image in terms of image fidelity considering the overall similarity, level of detail preservation, and visual quality. A higher rating indicates a closer resemblance to the original image, while a lower rating implies a greater deviation or degradation.

2. **Preservation of clinically-relevant features:** This question evaluates the extent to which the reconstructed chest X-rays image preserves the diagnostic information present in the original image given the clarity and visibility of anatomical structures, abnormalities, and other important diagnostic features. A higher rating indicates a

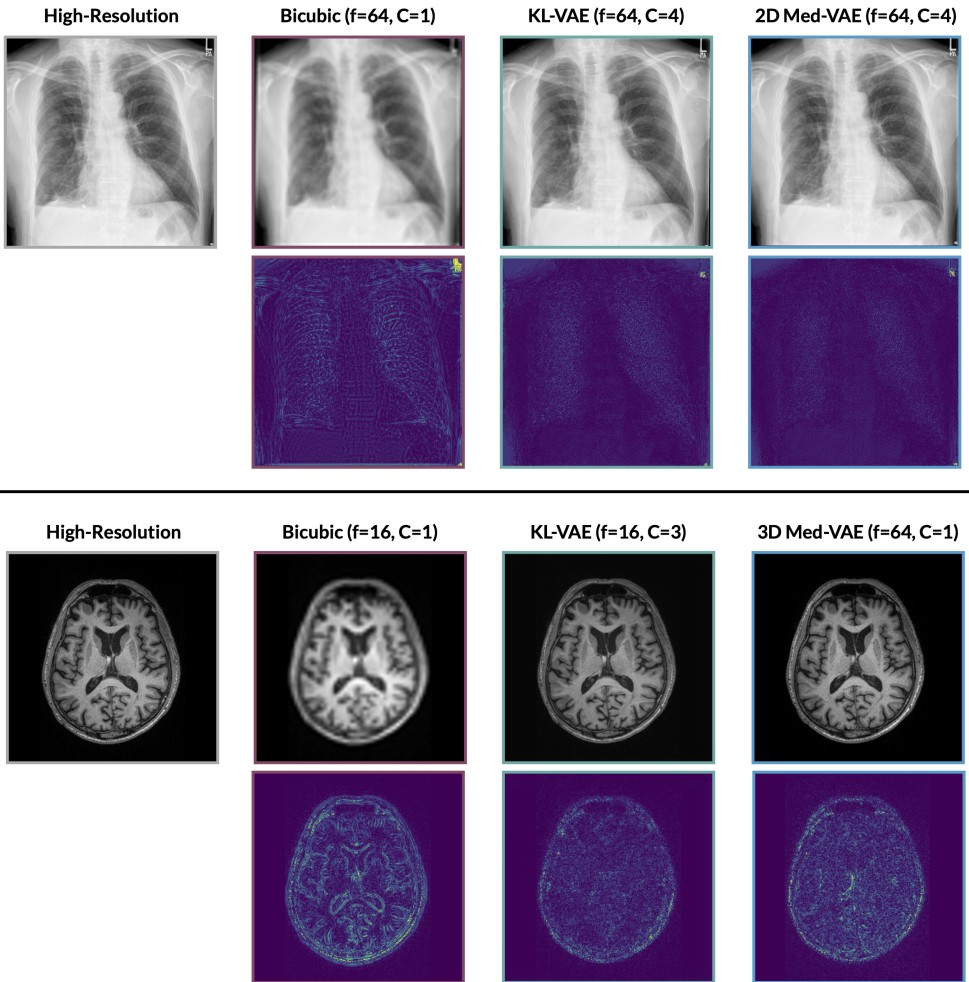

Figure 4: **Qualitative examples of reconstructed medical images.** The top section provides qualitative examples of a reconstructed chest X-ray. The bottom section provides qualitative examples of a reconstructed brain MRI slice. Residual figures show pixel-level differences between reconstructed images and original, high-resolution images; brighter colors represent larger differences.

greater preservation of diagnostic information, while a lower rating suggests a significant loss that may affect the accuracy of diagnosis.

3. **Presence of Artifacts:** This question focuses on the presence and impact of artifacts in the reconstructed chest X-ray. Artifacts can include image distortions, noise, blurring, or other visual anomalies (ie. hallucinations) that are not present in the original image. A higher rating suggests less or no interference from artifacts, while a lower rating suggests a greater occurrence of artifacts.

For automated perceptual quality evaluations on 2D images, we calculate PSNR and MS-SSIM on a random sample of 1000 images for each image type; we report mean and

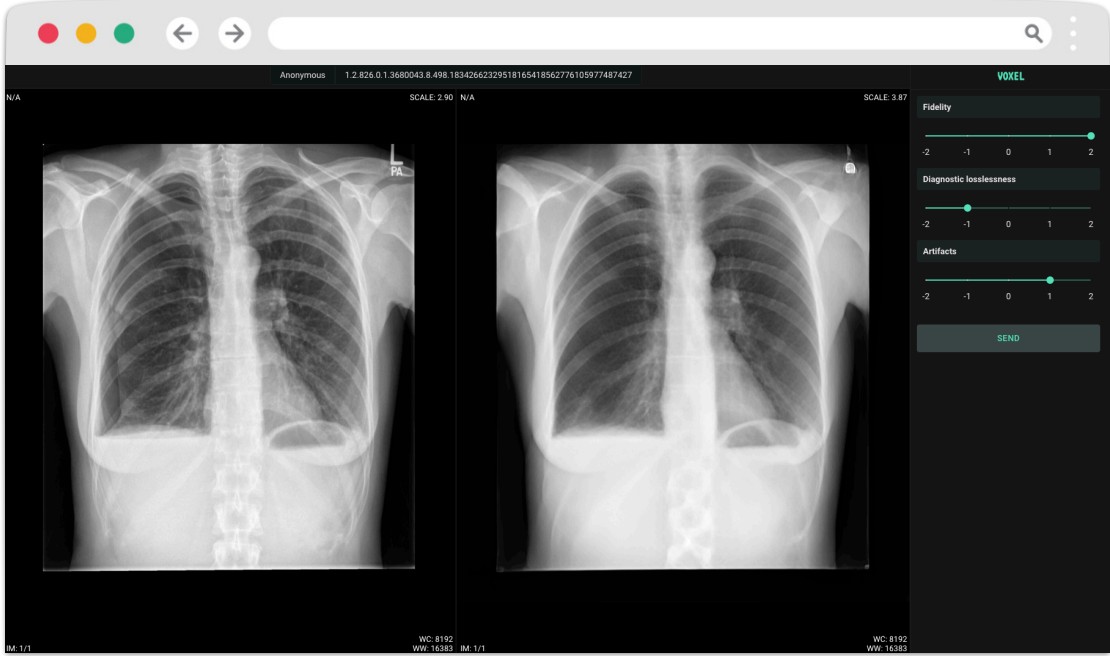

Figure 5: **Reader study user interface**. Expert readers score each reconstructed chest x-ray with respect to image fidelity, preservation of clinically-relevant features, and the presence of artifacts. Each expert reader is presented with a pair of chest X-rays, consisting of an original high-resolution image $x$ on the left and a reconstructed image $\hat{x}$ on the right. Readers are blinded to both the method and the downsizing factor used to generate the reconstructed image.

standard deviations across four runs with different random seeds. For automated perceptual quality evaluations on 3D images, we calculate PSNR and MS-SSIM on a single random sample of 100 images for each image type. For manual perceptual quality evaluations with expert readers, we report mean scores and 95% confidence intervals across three readers.

In Figure 4, we provide qualitative examples of reconstructed medical images.

## B.3. Ablations

We analyze the effects of each stage of training on latent representation quality in Table 8 and Table 9.

We further ablate the inclusion of the embedding consistency loss term in the Stage 1 training procedure. We find that the embedding consistency loss term helps improve reconstructed image quality, particularly at lower compression factors. For instance, at a compression factor of $f = 16$, Stage 1 training without the embedding consistency loss term achieves a PSNR of $37.27_{\pm 0.08}$ and an MS-SSIM of $0.992_{\pm 0.0}$ on mammograms. In comparison, Stage 1 training with the embedding consistency loss term achieves a PSNR of $37.57_{\pm 0.08}$ and an MS-SSIM of $0.993_{\pm 0.0}$, as shown in Table 2.

| Method | $f$ | $C$ | AUROC ↑ | | | | | |
|---|---|---|---|---|---|---|---|---|
| | | | Malignancy | Calcification | BI-RADS | Bone Age | Wrist Fracture | Avg. |
| High-Resolution | 1 | 1 | **66.1** | **62.4** | **63.4** | **80.2** | **73.7** | **69.2** |
| 2D Base Autoencoder (Stage 1) | 16 | 3 | 58.7 | 60.5 | 58.0 | 72.0 | 64.3 | 62.7 |
| 2D Med-VAE (Stage 2) | 16 | 3 | **66.1** | **61.7** | **62.3** | **82.1** | **70.6** | **68.6** |
| 2D Base Autoencoder (Stage 1) | 64 | 4 | 63.4 | 54.4 | 58.6 | 65.7 | 61.9 | 60.8 |
| 2D Med-VAE (Stage 2) | 64 | 4 | **64.9** | **58.5** | **60.6** | **73.0** | **66.7** | **64.7** |

Table 8: **Effect of each autoencoder training stage on 2D Med-VAE latent representation quality.** We evaluate the effects of each stage of 2D Med-VAE training on latent representation quality using five 2D CAD tasks.

| Method | $f$ | $C$ | AUROC ↑ | | | |
|---|---|---|---|---|---|---|
| | | | Spine Fractures | Skull Fractures | Knee Injury | Avg. |
| High-Resolution | 1 | 1 | **82.9** | **63.9** | **69.9** | **72.2** |
| 2D Base Autoencoder (Stage 1) | 64 | 1 | 76.1 | 36.6 | 65.0 | 59.2 |
| 3D Med-VAE (Stage 2) | 64 | 1 | **83.7** | **87.0** | **68.4** | **79.7** |
| 2D Base Autoencoder (Stage 1) | 512 | 1 | **72.5** | 45.4 | **68.8** | **62.2** |
| 3D Med-VAE (Stage 2) | 512 | 1 | 72.0 | **49.1** | 58.2 | 59.8 |

Table 9: **Effect of each autoencoder training stage on 3D Med-VAE latent representation quality.** We evaluate the effects of each stage of 3D Med-VAE training on latent representation quality using three 3D CAD tasks. Since Stage 1 training exclusively involves 2D images, we evaluate this model on 3D tasks by stitching 2D latent representations together across slices such that the size of the 2D latent representation matches those generated by 3D models.

## Appendix C. Extended Discussion

High-resolution medical images can result in large data storage costs and increased or intractable computational complexity for trained models. As the volume of data stored by hospitals continues to increase and large-scale foundation models become more commonplace, methods for inexpensively storing and efficiently processing high-resolution medical images become a critical necessity. In this work, we aim to address this need by introducing Med-VAE, a family of 6 large-scale autoencoders for medical images developed using a novel two-stage training procedure. Med-VAE encodes high-resolution medical images as downsized latent representations. We demonstrate with extensive evaluations that (1) downsized latent representations can effectively replace high-resolution images in CAD pipelines while maintaining or exceeding performance, (2) downsized latent representations reduce storage requirements (up to 512x) and improve downstream efficiency (up to 70x in model throughput) when compared to high-resolution input images, and (3) reconstructed images effectively preserve relevant features necessary for clinical interpretation by radiologists.

Several prior works have introduced powerful autoencoders capable of generating downsized latents for images. In particular, recent work on latent diffusion models has involved the development of several large-scale autoencoders, such as VQ-GANs and VAEs, trained on eight million natural images (Rombach et al., 2022; Kingma and Welling, 2013; Esser

et al., 2021; Krasin et al., 2017); downsized latents generated by these models were shown to capture relevant spatial structure as well as improve efficiency of downstream diffusion model training (Rombach et al., 2022). However, recent works have demonstrated that models trained on natural images often generalize poorly to medical images due to significant distribution shift (Guan and Liu, 2022; Van der Sluijs et al., 2023; Chambon et al., 2022), suggesting that existing natural image autoencoders may not be well-suited for the complexity of the medical image domain. Our evaluations on both latent representations and reconstructed images support this point, demonstrating that existing large-scale natural image autoencoders consistently underperform our domain-specific medical image autoencoders. These findings demonstrate the need for domain-specific models capable of understanding complex and fine-grained patterns across diverse imaging modalities and anatomical regions.

Our work aims to reduce computational costs associated with automated medical image interpretation by proposing the use of training datasets comprised of downsized Med-VAE latent representations rather than high-resolution medical images. For instance, given a chest X-ray training dataset with images of size $1024 \times 1024$ with 1 channel, our 2D Med-VAE model with $f = 64$ and $C = 1$ can generate downsized latent representations of size $128 \times 128$ with 1 channel, contributing to substantial downstream efficiency and storage benefits. We demonstrate with eight CAD tasks that latent representations do not result in the loss of clinically-important information; at a 2D downsizing factor of $f = 16$ and a 3D downsizing factor of $f = 64$, we observe equivalent or better performance than high-resolution images with substantial improvements over multiple existing downsizing methods. Med-VAE models can also generalize beyond the images included in the training set, as shown by performance on 2D musculoskeletal X-rays and 3D spine CTs. Importantly, the efficiency benefits of using latent representations are significant; in particular, using latent representations can contribute to large increases in batch sizes, which can be particularly useful in the modern era of self-supervised foundation models that rely heavily on the use of large batch sizes during training.

The Med-VAE autoencoder family includes two 3D autoencoders that are explicitly designed to downsize 3D medical imaging modalities (e.g. CT, MRI), a previously under-researched setting. Our results demonstrate that at a 3D downsizing factor of $f = 64$, the volumetric latent representations generated by 3D Med-VAE are substantially higher quality than those generated by stitching together 2D slices downsized using 2D baselines. This suggests that 3D autoencoders can better capture clinically-important volumetric patterns, such as fractures that span multiple slices. Efficiency benefits in the 3D setting are also notable, particularly since training downstream CAD models on high-resolution 3D volumes is often computationally expensive or intractable. At significantly higher downsizing factors ($f = 512$), we observe the benefits of 3D autoencoder training to be less pronounced, suggesting that users will need to carefully consider the tradeoffs between latent representation quality and desired downstream efficiency when selecting a Med-VAE model.

In addition to generating high-quality latent representations, Med-VAE models also include a trained decoder, which can reconstruct the original high-resolution image from the downsized latent. This is a particularly useful capability in the medical imaging domain, since high-resolution images are necessary for effective clinical interpretation by radiologists. We demonstrate with a reader study consisting of three radiologists that reconstructed

images can effectively preserve clinically-relevant signal needed for diagnoses; in this setting, fine-grained fractures in chest X-rays were preserved through the encoding and decoding process.

Our study presents several opportunities for future work. First, additional research into model architectures, data augmentation approaches, and training strategies would be useful for building effective downstream CAD models that can learn from latent representations. In addition, the batch size and efficiency benefits afforded by latent representations raise the possibility of training large-scale foundation models using downsized latent representations. Whereas foundation models traditionally require significant computational resources and training time, utilizing downsized latent representations that preserve diagnostic features can greatly accelerate model training, particularly in resource-constrained settings. Future work can explore foundation model performance and scaling laws in this context. Finally, future work can explore additional autoencoder training strategies to better preserve clinically-relevant features at high downsizing factors.

Overall, our work demonstrates the potential that large-scale, generalizable autoencoders hold in addressing critical storage and efficiency challenges in the medical domain.

