# OpenReview forum: "MedVAE: Efficient Automated Interpretation of Medical Images with Large-Scale Generalizable Autoencoders"
_MIDL.io/2025/Conference — MIDL 2025 Oral_

### Official Review · Reviewer_Kadx · 2025-02-21

**Confidence:** 3
**Preliminary Rating:** 4
**Recommendation:** Poster

**Summary:**

This paper presents Med-VAE, a family of 6 large-scale 2D/3D VAEs for encoding and reconstructing medical images. MedVAE is trained on over 1 million medical images, and evaluated across 8 different detection/prediction tasks over X-ray, CT, FFDM, and MRI data, demonstrating improvements over interpolation, KL-VAE, and VQ-GAN.

**Strengths:**

MedVAE is trained and tested on a large dataset with diverse tasks and modalities.

Evaluation of MedVAE considered a suite of relatively comprehensive metrics including detection/prediction, storage and efficiency, and reconstruction.

MedVAE can have broader impact for medical image research.

**Weaknesses:**

The baselines seem to be relatively naive — it is not clear why other large-scale autoencoders are not considered in the comparison.

The relation between the training and evaluation datasets is not clear.

The descriptions of MedVAE can be more detailed, especially in the 2nd-stage training.

It would have been good to see some examples of reconstruction.

**Detailed Comments:**

Please see above.

**Justification Of The Preliminary Rating:**

MedVAE represents a valuable step ahead in building large-scale models capable of extracting latent representation from medical images that can preserve downstream task performance and image reconstruction quality. The training and evaluation of MedVAE demonstrated a substantial amount of work considering diverse medical image modalities and tasks. The evaluation was comprehensive in the aspects of performance considered. Some details of the presented work can be clarified, and the baselines seem to be strengthen-able. The overall work as presented however can have broader impact to accelerate future medical image research and clinical applications.

**Questions To Address In The Rebuttal:**

Please clarify the relation between the data used to train and evaluation MedVAE.

Please clarify 2nd stage training.

Please justify the reason why more advanced large-scale autoencoders are used in comparison.

---

> ### Author Response · Authors · 2025-03-08
> **Response to Reviewer KADX (1/2)**
>
> We thank Reviewer KADX for reviewing our work and providing helpful feedback. We respond to specific comments below.
>
> **"The baselines seem to be relatively naive — it is not clear why other large-scale autoencoders are not considered in the comparison...Please justify the reason why more advanced large-scale autoencoders are [not] used in comparison."**
>
> We compare MedVAE with two state-of-the-art autoencoders trained on natural images, referred to as KL-VAE and VQ-GAN in the main text. These models are comparable baselines for MedVAE, since both were trained on large-scale datasets (consisting of 8 million natural images) and have equivalent downsizing factors as MedVAE. We show that both 2D and 3D MedVAE models show consistent improvements in performance over these comparable baselines, demonstrating the importance of domain-specific training.
>
> **"The relation between the training and evaluation datasets is not clear...Please clarify the relation between the data used to train and evaluation MedVAE."**
>
> After training MedVAE using our collected large-scale dataset, we perform evaluation using eight CAD tasks (which measure latent representation quality) as well as a series of perceptual quality evaluations (which measure reconstructed image quality). All metrics reported in the paper are derived from test sets associated with the evaluation tasks; there is no overlap between these test sets and the data used to train MedVAE. In Appendix Section B.1, we provide further details on how the train, validation, and test sets were designed for each CAD task. We also perform a series of evaluations on completely held-out datasets; for example, we demonstrate strong performance of 2D MedVAE latents on bone age classification and wrist fracture detection, despite the fact that no musculoskeletal x-rays are used during MedVAE training.
>
> **"The descriptions of MedVAE can be more detailed, especially in the 2nd-stage training...Please clarify 2nd stage training."**
>
> Thank you for this suggestion. Due to space limitations in the main text, our original submission provided extended details on model training and implementation in Appendix Section A.3. In particular, in the context of 2D imaging modalities, the second training stage takes the form of a lightweight fine-tuning procedure designed to maximize consistency in clinically-relevant features between the input image and the latent representation. Our key insight here is that image embeddings generated by BiomedCLIP can effectively capture clinically-relevant features in 2D medical images, suggesting its utility as a guidance mechanism during training. We freeze all parameters in the encoder and decoder of the VAE. During training, the input image $x_i$ is passed through the frozen VAE encoder to generate the latent representation $z_i$; then, $z_i$ is passed through a series of lightweight, trainable projection layers, which yield an output representation $\hat{z_i}$ with the same size as $z_i$. Let the function $b(\cdot)$ represent the BiomedCLIP embedding function. We optimize the projection layer weights using a domain-specific embedding consistency loss, which takes the form of an $L_2$ loss between $b(x_i)$ and $b(\hat{z_i})$. All downstream evaluations of latent representation quality are performed with the projected latent $\hat{z_i}$. We perform Stage 2 training using the curated 2D training dataset with one million images. Our procedure yields four 2D MedVAE autoencoders with various downsizing factors and number of latent channels.
>
> In the context of 3D imaging modalities (e.g. CT scans, MRIs), the second training stage involves lifting the 2D VAE architecture to 3D using a kernel centering inflation strategy; we then continue training with 3D images. We note here that using external 2D medical foundation models like BiomedCLIP to enforce feature consistency is inadequate for 3D settings. As a result, we instead implement a training procedure focused on maximizing perceptual similarity, analogous to 2D stage 1 training. We train the 3D autoencoders using random cubic patches of size $64 \times 64 \times 64$. The perceptual loss and the patch-based adversarial objective are calculated per-slice, with the final loss term computed as the mean across all slices in the volume. Following such a training strategy, a 3D MedVAE model with $f = 64$, $C = 1$, and input image $x_i$ of size $H \times W \times S \times 1$ would generate a latent representation $z_i$ of size $(H/4) \times (W/4) \times (S/4) \times 1$, downsizing the volume by 64x. We perform Stage 2 training using the curated dataset of 31,374 3D images. Our procedure yields two 3D MedVAE autoencoders across various downsizing factors.

---

> > ### Author Response · Authors · 2025-03-08
> > **Response to Reviewer KADX (2/2)**
> >
> > **"It would have been good to see some examples of reconstruction."**
> >
> > We provide qualitative examples of reconstructed medical images in Appendix Figure 4. The top portion of Figure 4 provides qualitative examples of a reconstructed chest X-ray, and the bottom portion of Figure 4 provides qualitative examples of a reconstructed brain MRI slice. We also provide residual figures to show pixel-level differences between reconstructed images and original, high-resolution images, with brighter colors representing larger differences.
> >
> >
> > We again thank Reviewer KADX for their review of our manuscript, and we hope that the above responses adequately address all concerns.

---

> ### Author Response · Authors · 2025-03-12
>
> Dear Reviewer KADX, Thank you very much for your helpful feedback and suggestions. We have carefully addressed all of your comments in our response. Since the discussion period will be ending in two days, we would greatly appreciate any updated feedback. Thank you again for your time.

---

> ### Comment · Reviewer_Kadx · 2025-03-18
>
> I'd like to thank the authors for carefully clarifying my previous questions, and I'm happy to raise my rating to Accept.

---

### Official Review · Reviewer_wTfe · 2025-02-22

**Confidence:** 4
**Preliminary Rating:** 5
**Recommendation:** Oral
**Final Rating:** 5

**Summary:**

The work introduces MedVAE models to encode 2D and 3D medical images into downsized latent representations and decoding the images back into high resolution images. Experiments have been performed on 20 different datasets on multiple modalities to demonstrate high efficiency and ability to decode the representations into high resolution images.

**Strengths:**

- The paper is well-written and readable.
- This is a large scale study which provides a family of models that could be used for a variety of downstream tasks.
- In depth experimental details are shared in the appendix and code is available.
- Various metrics show the superiority of the approach against other methods. Evaluations on both latent representations and reconstructed images along with performance on downstream tasks have been reported.
- Ablation studies have been performed to highlight the impact of downsampling factor and number of channels.
- Manual validation has been performed by 3 radiologists.

**Weaknesses:**

- While the study is quite comprehensive, the explanations on the choice of image downsizing methods could be more elaborate.
- In the manual evaluations, the reason for excluding KL-VAE and VQ-GAN is not clear.

**Detailed Comments:**

- It would be good to have an analysis on the impact of stage 2 training on the framework i.e how much impact do BiomedCLIP embeddings have on the performance and potentially a qualitative analysis to highlight the implications.
- The specific endpoint used for BiomedCLIP in the study should be mentioned.

**Justification Of The Final Rating:**

This large scale study would be useful for the medical imaging domain. Experiments have been performed rigorously and manual validation adds value to the contribution. Overall, the contribution is meaningful and enables future research. My minor concerns have been addressed by the authors with additional minor updates to the manuscript. Therefore, I do not change my rating.

**Justification Of The Preliminary Rating:**

This large scale study would be useful for the medical imaging domain. Experiments have been performed rigorously and manual validation adds value to the contribution. Overall, the contribution is meaningful and enables future research.

**Questions To Address In The Rebuttal:**

In Table 4 of Appendix B.1 of the paper, is there any specific observation made on the 2D Med-VAE (f=512, C=4) performance on Skull Fractures? It may be good to highlight some edge cases and mention limitations/areas that can be addressed in future research in a separate section.

**Special Issue:**

Yes

---

> ### Author Response · Authors · 2025-03-08
> **Response to Reviewer WTFE (1/2)**
>
> We thank Reviewer WTFE for reviewing our work and providing helpful feedback. We respond to specific comments below.
>
> **“While the study is quite comprehensive, the explanations on the choice of image downsizing methods could be more elaborate.”**
>
> Thank you for your suggestion and for noting the comprehensiveness of our study. Our work compared MedVAE with two categories of downsizing methods: (1) interpolation methods and (2) large-scale natural image autoencoders, selected for the following reasons:
> - Interpolation Methods: Prior computational models trained on high-resolution medical images generally downsize images using interpolation methods, which are provided as part of standard machine learning packages including PyTorch and OpenCV. We select interpolation methods for comparison with MedVAE since interpolation serves as the current gold standard for medical image downsizing, as demonstrated by the quantity of prior work leveraging this approach. Interpolation methods generate compressed images $z_i$ by replacing each pixel with an aggregated value of sampled neighboring pixels. We benchmark three representative methods: nearest-neighbor, bilinear, and bicubic interpolation.
> - Large-scale natural image autoencoders: We select large-scale natural image autoencoders for comparison with MedVAE since recent work has demonstrated that large-scale neural compressors trained on millions of natural images can effectively compress images into downsized latent representations while preserving key visual features, leading to improvements in downstream computational efficiency [1]. We evaluate two recently-introduced neural compressors: KL-VAE and VQ-GAN. The KL-VAE compressors take the form of standard convolutional VAEs trained using a perceptual loss, a patch-based adversarial objective, and a penalty based on the Kullback-Leibler (KL) divergence that pulls latents towards a standard normal. The VQ-GAN compressors use a perpetual loss and a patch-based adversarial objective with a quantization step included prior to the decoder. Both versions were trained using eight million natural images collected from OpenImages.
>
> We have updated the manuscript with a related works section that includes these explanations.
>
> [1] Robin Rombach, et al. High-resolution image synthesis with latent diffusion models. In Proceedings of the IEEE/CVF Conference on Computer Vision and Pattern Recognition, pages 10684–10695, 2022
>
> **“In the manual evaluations, the reason for excluding KL-VAE and VQ-GAN is not clear.”**
>
> Our manual evaluations were specifically designed to compare interpolation methods to MedVAE because interpolation methods are the predominant gold-standard in the medical imaging literature; a vast majority of existing models designed for medical imaging settings utilize interpolation methods in order to perform image downsizing. Comparisons with other learned autoencoder-based approaches are performed via automated metrics in order to capture fine-grained subtleties that may be missed by the human eye. We also provide qualitative examples of reconstructed medical images across baselines (with associated residual figures) in Appendix Figure 4.
>
> **“It would be good to have an analysis on the impact of stage 2 training on the framework i.e how much impact do BiomedCLIP embeddings have on the performance and potentially a qualitative analysis to highlight the implications.”**
>
> Thank you for this suggestion. In our original submission, Table 8 (in Appendix Section B.3) analyzed the effects of each stage of 2D MedVAE training on latent representation quality. In response to this suggestion, we have updated the main text of the manuscript to reference this ablation.
>
> As shown in Table 8, our second training stage with BiomedCLIP leads to considerable improvements in performance across all five evaluated 2D CAD tasks. Specifically, at a downsizing factor of $f=16$, we observe a 9.4% improvement in model performance on average after the incorporation of the second training stage. Similarly, at a downsizing factor of $f=64$, we observe an average performance improvement of 6.4%. We note that the second training stage for 2D MedVAE solely focuses on improving latent representation quality and does not affect reconstructions (Figure 1b); as a result, qualitative analyses and quantitative metrics characterizing reconstruction quality will be equivalent across both stages.
>
> **“The specific endpoint used for BiomedCLIP in the study should be mentioned.”**
>
> We have updated our manuscript (Appendix Section A.3) to clarify that we use the BiomedCLIP-PubMedBERT_256-vit_base_patch16_224 model available on HuggingFace at https://huggingface.co/microsoft/BiomedCLIP-PubMedBERT_256-vit_base_patch16_224.

---

> > ### Author Response · Authors · 2025-03-08
> > **Response to Reviewer WTFE (2/2)**
> >
> > **“In Table 4 of Appendix B.1 of the paper, is there any specific observation made on the 2D Med-VAE (f=512, C=4) performance on Skull Fractures? It may be good to highlight some edge cases and mention limitations/areas that can be addressed in future research in a separate section.”**
> >
> > Thank you for this question. We note variability in MedVAE performance on the skull fracture task at the 512x compression level as detailed in Table 4 (Appendix B.1). Specifically, 2D MedVAE (f=512, C=4) resulted in a lower AUROC ($17.1_{\pm 8.6}$) compared to its single-channel 2D and 3D MedVAE counterparts ($63.0_{\pm 1.1}$ and $49.1_{\pm 19.8}$ respectively). We attribute this variability to the significant compression factor of 512x especially for medical images. For instance, a skull fracture in a head CT volume may have anisotropic dimensions of $8 \times 8 \times 4$. In the case of 512x compression, the skull fracture in the latent would become $1 \times 1 \times 1/2$, which means that the fracture feature may be smaller than an observable voxel. As a result, the latent classifier for skull fracture must attempt to separate and interpret partial-volume effects. This task is inherently challenging and can contribute to more pronounced performance variability.
> >
> > We highlight that at lower compression levels of 64x, our results show higher performance of 3D MedVAE ($87.0_{\pm 7.3}$) compared to interpolation, natural image autoencoders, and  2D MedVAE ($57.4_{\pm 4.0}$ and $50.9_{\pm 19.5}$ respectively) baselines. In this compression regime, the same $8 \times 8 \times 4$ skull fracture would now be expressed over $2 \times 2 \times 1$ voxels, which means that the classifier would not need to account for partial volume effects, thus allowing it to focus on extracting fracture features from the CT scans.
> >
> > For further discussion on these limitations and potential strategies for improvement, we refer the reviewer to our extended discussion in Appendix C, where we mention that "future work can explore additional autoencoder training strategies to better preserve clinically relevant features at high downsizing factors."
> >
> > We again thank Reviewer WTFE for their review of our manuscript, and we hope that the above responses adequately address all concerns.

---

> > > ### Comment · Reviewer_wTfe · 2025-03-14
> > >
> > > I thank the authors for actively engaging during the discussion period, providing detailed response to my concerns and making updates accordingly.
> > >
> > > With respect to the performance drop in skull fractures task, I understand the reasoning behind the drop due to the downsampling/compression factor. However, there is a significant drop in the 4-channel 2D Med-VAE (f=512) as compared to 1-channel 2D Med-VAE (f=512). The edge case could be useful for future development in case any interesting and useful insights are drawn here.

---

### Official Review · Reviewer_PTWL · 2025-02-23

**Confidence:** 5
**Preliminary Rating:** 4
**Recommendation:** Oral
**Final Rating:** 5

**Summary:**

-authors presented MedVAE, a family of six large-scale 2D and 3D autoencoders designed to encode medical images into compact latent representations, significantly improving computational efficiency while preserving clinically relevant features.
-Trained on 1,052,730 medical images across 20 datasets, MedVAE achieves up to 70x throughput improvement for downstream tasks and accurately reconstructs high-resolution images, demonstrating the potential of generalizable autoencoders to address efficiency challenges in medical AI.
-Large-scale experiments with very promising results.

**Strengths:**

strengths:
+A novel two stage training scheme, designed to optimize the quality of latent space and decoded reconstruction.
+more than 1m images were used for training, >=20 dataset,
+adaptation of 2D architectures to 3D ones.
+demonstrated a potential of generalizable auto encoders.
+clear writing, smooth flow, promising results, good rigor.
+code is there (did not check model files yet)

**Weaknesses:**

-results do not seem very high from the existing ones. I am a bit surprised after sing large scale training (promising) but not so significantly different results (in many experiments, similar results...or close).
-latent space compression for efficiency is not entirely new (minor comment)
-ablation studies could be stronger (impact of perceptual loss vs BiomedCLIP embeddings etc)

**Detailed Comments:**

My comments in weaknesses part, I believe, are self-descriptive. Please see that part.

**Justification Of The Final Rating:**

Authors clarified the unclear points, and did extra comparison and experiments.
Their responses sound logical and convincing.
I think, overall it is a good paper with interesting and promising results.

**Justification Of The Preliminary Rating:**

This paper introduces MedVAE, a large scale autoencoders designed for medical images, which can encode high-resolution images into downsized latent representations and decode them back. The key contribution is a two-stage training scheme that optimizes both the quality of latent representations and the fidelity of decoded reconstructions. MedVAE aims to address storage and computational efficiency challenges in the medical domain while preserving clinically relevant features. Despite some minor and moderate weaknesses, I think the paper is in high quality and results are interesting to discuss in the community.

**Questions To Address In The Rebuttal:**

-results do not seem very high from the existing ones. can you please comment why?
-latent space compression for efficiency is not entirely new, can you discuss some other compression methods to locate your methods among them.
-ablation studies could be stronger , any chance if you can search the impact of perceptual loss vs BiomedCLIP embeddings etc?

**Special Issue:**

Yes

---

> ### Author Response · Authors · 2025-03-08
> **Response to Reviewer PTWL (1/2)**
>
> We thank Reviewer PTWL for reviewing our work and providing helpful feedback. We respond to specific comments below.
>
> **"Results do not seem very high from the existing ones. can you please comment why?"**
>
> While our evaluations show that MedVAE achieves substantial performance improvements over existing approaches, two key considerations should be taken into account when interpreting the quantitative results. First, we compare MedVAE with a series of strong baselines (namely, interpolation methods and natural image autoencoders). Interpolation methods are the predominant gold-standard for downsizing images in the field of medical imaging, and the natural image autoencoders utilized in this work are trained on over eight million images and have been extensively validated in prior work. Second, we note that for our latent representation evaluations (Section 4.1), the original high-resolution images establish an implicit upper bound on maximum achievable performance; any evaluated downsizing method will only be capable of achieving marginal improvements past this bound. Regardless of this constraint, we demonstrate that MedVAE is capable of exceeding this bound on four out of five 2D classification tasks as well as two out of three 3D classification tasks, suggesting that MedVAE effectively preserves clinically-relevant features.
>
> **"Latent space compression for efficiency is not entirely new, can you discuss some other compression methods to locate your methods among them."**
>
> Thank you for this suggestion. Our original submission included an extended discussion in Appendix Section C that discussed related works. In response to this suggestion, we have inserted a related works section in the main text in order to better situate our approach among related works.
>
> Prior computational models trained on high-resolution medical images generally downsize images using interpolation methods, which are provided as a part of standard machine learning packages including PyTorch and OpenCV. However, such approaches have been shown to result in degraded performance, raising the possibility of using learned approaches, such as autoencoders (Sabottke and Spieler, 2020b). Autoencoders are classic machine learning models consisting of encoders, which encode input images as downsized latent representations, and decoders, which decode latent representations back to the pixel space (Kingma and Welling, 2013; He et al., 2021; Vincent et al., 2008). In this work, we specifically focus on autoencoders that generate structured latent representations interpretable as downsized images. We note here that many standard autoencoders instead yield latent representations that take the form of vectors (He et al., 2021; Zhou et al., 2023); we consider these models out of scope for this work, since resulting latents cannot be used as drop-in replacements for images in CAD pipelines and will require downstream architecture modifications (particularly for fully-convolutional CAD models).
>
> Several recent works have introduced powerful autoencoders capable of generating downsized latents for images. In particular, recent works on latent diffusion models have involved the development of several large-scale autoencoders, such as VQ-GANs and VAEs, trained on eight million natural images (Rombach et al., 2022; Kingma and Welling, 2013; Esser et al., 2021; Krasin et al., 2017); downsized latents generated by these models were shown to capture relevant spatial structure as well as improve efficiency of downstream diffusion model training (Rombach et al., 2022; Chambon et al., 2022a). However, recent works have demonstrated that models trained on natural images often generalize poorly to medical images due to significant distribution shift (Guan and Liu, 2022; Van der Sluijs et al., 2023; Chambon et al., 2022b); as a result, it is unclear whether existing natural image autoencoders can effectively capture fine-grained clinically-relevant features. In the medical domain, existing works have primarily focused on developing autoencoders for small, focused sets of imaging modalities and anatomical features (e.g. chest X-rays) (Lee et al., 2023; Pinaya et al., 2022). In contrast, our work focuses on the development of large-scale, generalizable autoencoders capable of operating on diverse features, anatomical regions, and modalities (including 2D and 3D images). We also conduct evaluations to rigorously assess the quality of generated latent representations and reconstructed images in the medical domain.

---

> > ### Author Response · Authors · 2025-03-08
> > **Response to Reviewer PTWL (2/2)**
> >
> > **"Ablation studies could be stronger, any chance if you can search the impact of perceptual loss vs BiomedCLIP embeddings etc?"**
> >
> > Thank you for this suggestion. In our original submission, Appendix Section B.3 provided ablations analyzing the effects of each stage of MedVAE training on latent representation quality. We have (1) updated the main text of the manuscript to reference this ablation and (2) added a new ablation analyzing the inclusion of the embedding consistency loss. We discuss these ablations in further detail below.
> >
> > As shown in Table 8, our second training stage with BiomedCLIP leads to considerable improvements in performance across all five evaluated 2D CAD tasks. For example, at a downsizing factor of $f=16$, we observe a 9.4% improvement in model performance on average after the incorporation of the second training stage. In Table 9, we provide a similar ablation with our 3D MedVAE models, demonstrating the utility of our second training stage, particularly at lower downsizing factors (i.e. we observe a 34.6% average performance improvement at a downsizing factor of $f=64$).
> >
> > We further ablate the inclusion of the embedding consistency loss term in the Stage 1 training procedure. We find that the embedding consistency loss term helps improve reconstructed image quality, particularly at lower downsizing factors. For instance, at a downsizing factor of $f=16$, Stage 1 training without the embedding consistency loss term achieves a PSNR of $37.27_{\pm 0.08}$ and an MS-SSIM of $0.992_{\pm 0.0}$ on mammograms. In comparison, Stage 1 training with the embedding consistency loss term achieves a PSNR of $37.57_{\pm 0.08}$ and an MS-SSIM of $0.993_{\pm 0.0}$, as shown in Table 2.
> >
> > We again thank Reviewer PTWL for their review of our manuscript, and we hope that the above responses adequately address all concerns.

---

> > > ### Comment · Reviewer_PTWL · 2025-03-08
> > > **I found the responses inclusive and complete**
> > >
> > > Thank you for taking a serious time and updating the manuscript, and include also discussion points and answers what I have asked.
> > >
> > > I will increase my score even though it was already an accept level.
> > >
> > > thanks for the good work!

---

### Comment · Reviewer_PTWL · 2025-02-23
**large-scale experiments with very promising results.**

This paper introduces MedVAE, a large scale autoencoders designed for medical images, which can encode high-resolution images into downsized latent representations and decode them back. The key contribution is a two-stage training scheme that optimizes both the quality of latent representations and the fidelity of decoded reconstructions. MedVAE aims to address storage and computational efficiency challenges in the medical domain while preserving clinically relevant features.

strengths:
+A novel two stage training scheme, designed to optimize the quality of latent space and decoded reconstruction.
+more than 1m images were used for training, >=20 dataset,
+adaptation of 2D architectures to 3D ones.
+demonstrated a potential of generalizable auto encoders.
+clear writing, smooth flow, promising results, good rigor.
+ code is available (did not check all the details yet)

weaknesses:
-results do not seem very high from the existing ones. I am a bit surprised after sing large scale training (promising) but not so significantly different results (in many experiments, similar results...or close).
- latent space compression for efficiency is not entirely new (minor comment)
- ablation studies could be stronger (impact of perceptual loss vs BiomedCLIP embeddings etc)


Decision:
I would go for "accept" for this paper with minor correction from weaknesses.

---

### Author Rebuttal · Authors · 2025-03-08

**Rebuttal:**

We thank the reviewers for their thoughtful review of our manuscript. We were encouraged to see that all reviewers assigned an "accept" rating to our work. Reviewers noted that the paper is "well-written" (Reviewers PTWL, WTFE); the proposed method is "novel" (Reviewer PTWL); the experiments demonstrate "good rigor" with "in-depth details" (Reviewers PTWL, WTFE); and the proposed models hold "broader impact for medical image research" (Reviewer KADX).

In this work, we introduced MedVAE, a family of six large-scale 2D and 3D autoencoders capable of encoding medical images as downsized latent representations and decoding latent representations back to high-resolution images. Our key aim in developing MedVAE is to address the challenge of downsizing medical images in order to improve downstream computational efficiency while preserving clinically-relevant features. We train MedVAE autoencoders using a novel two-stage training approach with 1,052,730 medical images. Across diverse tasks obtained from 20 medical image datasets, we demonstrate that (1) utilizing MedVAE latent representations in place of high-resolution images when training downstream models can lead to efficiency benefits (up to 70x improvement in throughput) while simultaneously preserving clinically-relevant features and (2) MedVAE can decode latent representations back to high-resolution images with high fidelity.

In response to feedback, we provide individual responses below to address each reviewer’s concerns as well as an updated manuscript (with changes highlighted in red). We would again like to thank all reviewers for their time and feedback, and we hope that our changes adequately address all concerns.

**Supporting Material:**

/attachment/ca98514cbe23ff53bdd1c6c539c225646b950976.pdf

---

### Meta-Review · Area_Chair_KTf2 · 2025-03-16

**Recommendation:** Accept (Oral)
**Confidence:** 5

**Metareview:**

This work explores the efficient and compact representation of medical images through large-scale 2D and 3D variational autoencoders, aiming to achieve computational efficiency while preserving clinically relevant features. The database used for both training and testing comprises more than one million images, which is exceptional. A key methodological contribution lies in the use of a two-stage training approach leveraging the embedding space of BiomedCLIP to compute a domain-specific loss. An ablation study confirmed the relevance of this approach. One limitation, however, is the choice of placing the methodological description in the appendices. The discussion phase has been particularly productive, leading to a consensus among reviewers on the quality of this work.

For all these reasons, I have decided to accept this article. I would also like to congratulate the authors on the high quality of their work.